# Transformed Shell Roof Structures as the Main Determinant in Creative Shaping Building Free Forms Sensitive to Man-Made and Natural Environments

**Jacek Abramczyk**

Department of Architectural Design and Engineering Graphics, Rzeszow University of Technology, Al. Powstańców Warszawy 12, 35-959 Rzeszów, Poland; jacabram@prz.ed.pl; Tel.: +48-795-486-426

**Abstract:** The article presents author's propositions for shaping free forms of buildings sensitive to harmonious incorporation into built or natural environments. Complex folded structures of buildings roofed with regular shell structures are regarded as the most useful in creative shaping the free forms that can easily adapt to various expected environmental conditions. Three more and more sophisticated methods are proposed for creating variously conditioned free form structures. The first method allows the possibility of combining many single free forms into one structure and leaves the designer full freedom in shaping regular or irregular structures. The second, more sophisticated method introduces additional rules supporting the designer's spatial reasoning and intuition in imposing regularity of the shapes of the building structure and its roof shell structure. The third, most sophisticated method introduces additional conditions allowing the optimization of the regular shapes and arrangement of complete shell roof segments on the basis of an arbitrary reference surface and a finite number of straight lines normal to the surface. This original, interdisciplinary study offers new insight into, and knowledge of, unconventional methods for the creative shaping of innovative free forms, where great possibility and significant restrictions result from geometrical and mechanical properties of the materials used. Solving a number of issues in the field of civil engineering, descriptive geometry and architecture is crucial in the process of creating these structures.

**Keywords:** building free form structure; corrugated shell roof; integrated architectural form; thin-walled open profile; shape transformation; folded sheet

## 1. Introduction

Curved metal shell roofs have been used since the Gothic and became very popular in the Renaissance owing to their attractive architectural forms and stable constructions [1–3]. Glass and laminated glass elements made of reinforced polymers are used as structural members together with metal ones, which diversifies and improves the attractiveness of the architectural forms of buildings [4–6]. Space grids and complete shells are combined into a single internal coherent shell structure to strengthen the shell roofs and improve their stability [7–10].

Open thin-walled steel sheets folded in one direction joined with their longitudinal edges into flat sheeting can be easily transformed into shell forms as a result of assembling them to skew roof directrices [11,12]. The shell shape of the sheeting depends on a mutual position and curvature of the directrices and can be modelled with the help of warped surface [13]. The transformations are effective if freedom of the transversal width increments of each shell fold at its length is ensured to obtain positive static-strength work [14]. Such transformed sheeting is characterized by big mutual displacements of its subsequent folds in the shell, small strain and big deformations of the fold's flanges and webs [14,15] (Figure 1).

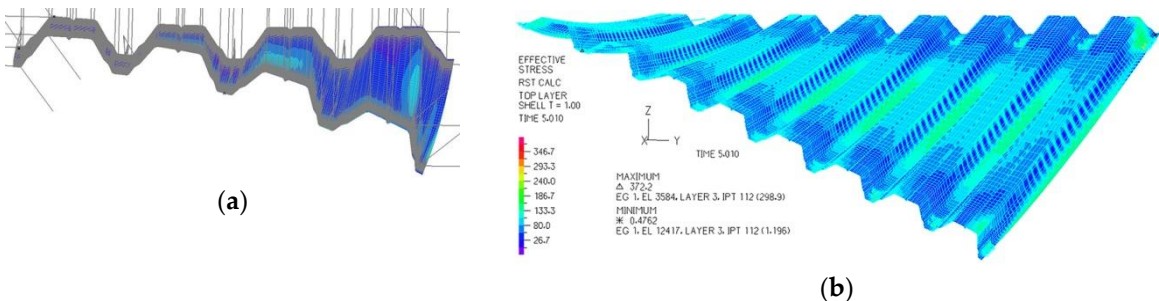

**Figure 1.** Accurate thin-walled folded computer models: (**a**) a nominally plane folded sheet transformed into a shell shape; (**b**) nominally plane folded sheeting transformed into a shell shape and loaded with a characteristic load.

Because of the above big displacements and effective shape transformations, great freedom in shaping the roof shell forms is achieved by means of two directrices adopted almost freely, so a variety of the architectural free forms of the resultant shell roofs and entire buildings is great [9]. Some important geometrical and mechanical restrictions of the sheet's shape transformations have to be taken into account. The basic one concerns the fact that each effectively transformed fold contracts at its half-length and is stretched at both crosswise ends [16]. Therefore, two or more complete corrugated shell sheets cannot be joined with their crosswise ends, that is perpendicular to the fold's directions, to obtain one resultant smooth shell [17]. They can only be set together with their transverse ends (Figure 2), to obtain an edge roof shell structure with regular edge pattern on its surface [18,19]. For engineering developments, each shell fold can be modelled with a simplified smooth sector of a warped surface [20,21] including hyperbolic paraboloid [22,23]. The sum of all such sectors is a model of a continuous edge structure [13].

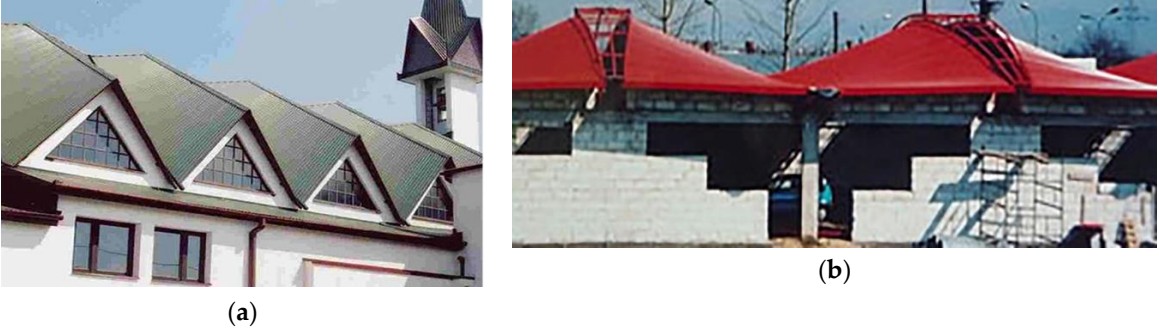

**Figure 2.** Two shell roof structures characterized by: (**a**) two straight directrices; (**b**) straight and curved directrices.

The considered transformed shells are stiffened with roof directrices transversally in relation to the fold's directions, and additional edge elements in order to maintain the straightness of the border folds in the shell [11–13]. Therefore, the considered structures need respective shapes of stiffened structural systems [7,8,11].

## 2. Critical Analysis

The use of well-known conventional design methods [6–10,21], known from the traditional courses of theory of structures, in the shaping of shell roof forms is ineffective because it usually results in high values of normal and shear stresses, local buckling and distortion of thin-walled flanges and webs of transformed shell folds. The assembly of the designed shell sheeting into skewed roof directrices is often impossible because of the plasticity of the fold's edges between flanges and webs. Reichhart developed a specific method for calculating the arrangement and the length of the supporting lines of

all folds in transformed corrugated shell sheeting [11], but it is effective only for the cases where the fold's longitudinal axes are perpendicular to roof directrices or very close to those [13]. The author significantly improved the Reichhart concept and has proposed an innovative method [13,18], so that the transformation would cause the smallest possible initial stresses on the shell folds resulting from this transformation.

For effective fold transformations, interdependence between the geometrical supporting conditions and the obtained shell forms of a transformed fold in a shell can be used [11]. In these cases, the freedom of the transverse width and height increments of each shell fold forming the transformed sheeting is ensured, and various attractive and innovative shapes of shell roofs and contraction curves of relatively big curvatures on these roofs can be achieved (Figure 3) [13]. If the fold does not have the freedom of transverse width increments due to strong stiffening of its longitudinal edges shared with its adjacent folds, the aforementioned interdependence cannot be used. Neither can it be used if the assembly technique causes additional forces varying the effective widths of the fold ends and their supporting lines.

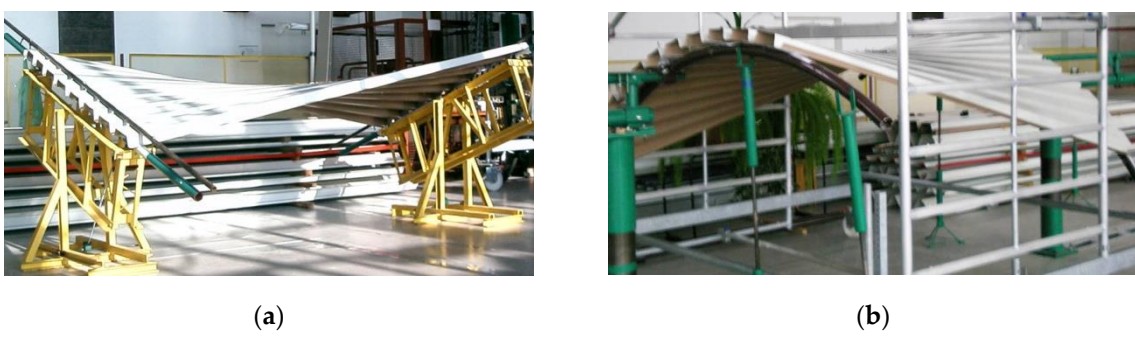

(**a**)                                          (**b**)

**Figure 3.** Experimental transformed corrugated shells supported by: (**a**) straight directrices, (**b**) curved directrices.

In the 1970s, Gergely, Banavalkar and Parker [24] accomplished shape transformations of folded sheets to create shallow right hyperbolic paraboloid roofs and their structures, named "hypars". The very limited shapes of shell roof structures using various configurations of hypars units are also discussed by Bryan and Davies [25]. Right hyperbolic paraboloids are a specific kind of hyperbolic paraboloids whose two rulings belonging to various families of rulings are perpendicular to each other. These two rulings are various lines of contraction of each right hyperbolic paraboloid [13]. Quarters and halves of these central sections are also used and joined together to obtain various shell structures, including hypars [24–26] (Figure 4).

The methods proposed by these authors drastically limit the variety of the designed transformed folded shell forms to central sectors of right hyperbolic paraboloids [26] and their one-fourths [24,25]. Moreover, the models obtained by means of these methods enforce unjustified additional stresses of the folds resulting from the need to adjust the longitudinal axes of the shell folds to the positions of selected rulings of the hyperbolic paraboloid used. The above adjustment of the longitudinal shell fold's axes to these rulings imposes a significant change in the width $\Delta M$ (Figure 5) [11] of the transverse fold's ends passing along shell directrix *LM*. These additional forces cause a significant increase in initial stresses and limitation of the searched surfaces to shallow hypars.

Simple shell structures composed of a few corrugated shells have been used in different architectural configurations, most often as shells supported by stiff constructions based on very few columns [27,28]. Shell structures are used for achieving: (a) large spans; (b) greater architectural attractiveness; and (c) skylights letting sunlight into the building interior [29,30].

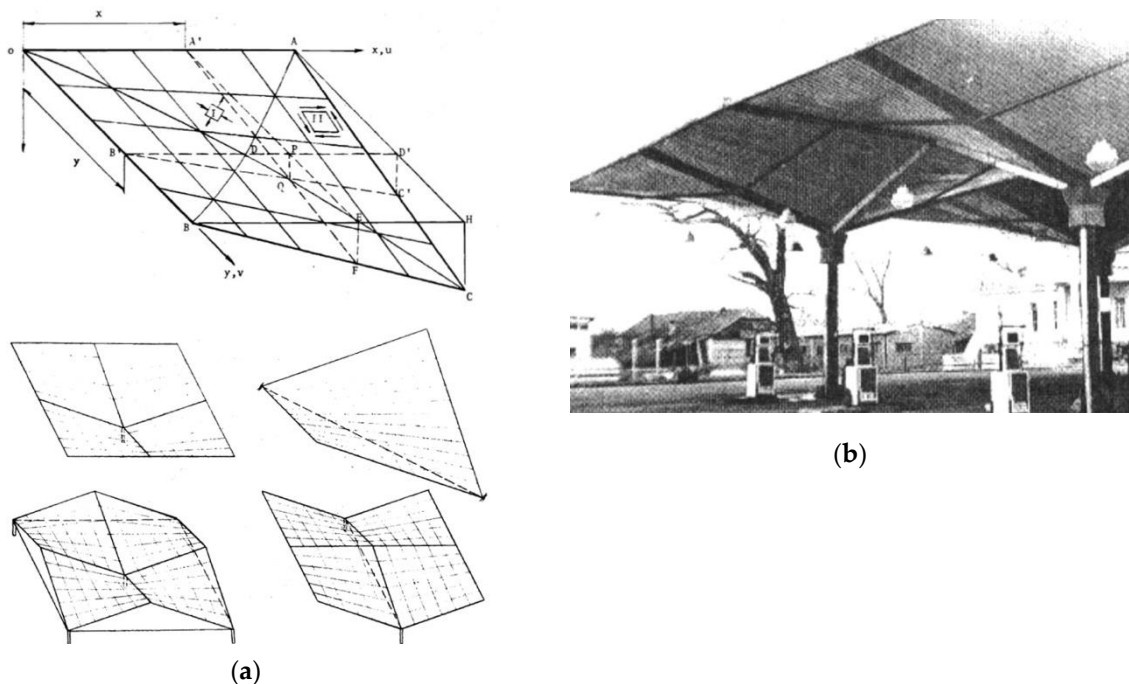

**Figure 4.** Roof shell structures (**a**) geometrical models; (**b**) erected construction.

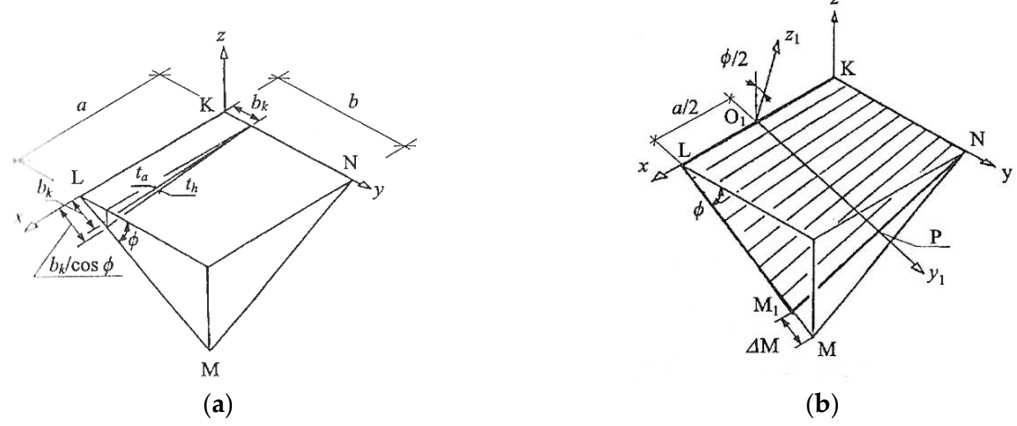

**Figure 5.** Adjustment of the longitudinal shell fold's axes to the selected rulings of one-fourth of the right hyperbolic paraboloid forcing significant change in the width $\Delta M$ of the transverse fold's ends passing along shell directrix *LM:* (**a**) forced change of transverse edge of individual transformed sheet, (**b**) forced change of transverse edge of complete transformed folded shell.

There are very few methods for the geometrical shaping of folded steel roofs transformed into shell forms. Among them, only the Reichhart method allows complete shells different from the central sectors of right hyperbolic paraboloids to be obtained [11]. In the 1990s, Reichhart started to shape corrugated steel sheeting for shell roofing, where all folds underwent big transformations into shell shapes. An additional advantage of the Reichhart method is that the initial stresses induced by the shape transformations are the smallest possible. Reichhart called such transformations free deformations, because they assure freedom of the transversal width and height increments of all folds in the transformed shell. In this way, the initial fold's effort is reduced to a possibly low level.

Reichhart arranged the complete corrugated shells on horizontal or oblique planes [11] as continuous ribbed structures (Figure 2). He developed a simple method for geometrical and strength

shaping of the transformed shell roofs. He designed corrugated shell sheeting supported by very stiff frameworks or planar girders with additional intermediate members and roof bracings. [20].

The transverse ends of transformed folds cannot be extended to the positions predicted by conventional methods, because this action causes a radical increase in stresses of the deformed thin-walled profiles [11,13,14]. The fold's fixing points along all roof directrices must be precisely calculated either by the Reichhart method [11], if the longitudinal fold's axes are close to perpendicular to the directrices, or by the author's method [13,18]. For these calculations based on precisely calculated supporting conditions, diversified for the subsequent folds in a roof shell, and the stiffness of these folds resulting from their geometrical and mechanical properties can be applied.

Unfortunately, the Reichhart method is correct only when the longitudinal axes of the transformed shell folds are perpendicular to the roof directrices, or very close to those, and the algebraic equations of these directrices are of the second order at most. In other cases, the method leads to serious errors, as demonstrated by the author [13]. These errors result from the lack of conditions providing similar values of stress at both transverse ends of the same fold. The visible result of different stress values at both transverse ends of the same shell fold is that the transverse contraction of the fold does not pass halfway along its length, on the contrary, it is shifted closer to one of these ends. The condition defined by the author is employed in his innovative method of shaping individual roof shells (monograph) [13] and implemented in the application he developed [18] in the Rhino/Grasshopper program used for parametric modeling of engineering objects.

The author started with experimental tests [13] and computer analyses [14] on static and strength work of folded sheets transformed into various shell forms and structural systems dedicated to supporting the transformed complete roof shells and their complex structures. This issue goes beyond the scope of the article. As Reichhart incorrectly accepted each shell fold as prismatic beams not cooperating with each other and having linear geometrical and mechanical characteristics, the author began his work with a preliminary understanding of the geometrical and static-strength characteristics of thin-walled folded shells. He analyzed the possibilities of modeling the transformed steel sheeting with accurate, thin-walled, folded computer models created in the ADINA program used for advanced dynamic incremental nonlinear analyses [15] (Figure 1). In order to accurately configure his computational models, he intends to perform experimental tests in the near future on the innovative experimental stand of his design at a laboratory hall (Figure 3).

The present article concerns geometrical shaping the building's free forms roofed with folded steel sheeting transformed into various shell forms [31]. Therefore, the possibilities of shaping complex free forms, that is free form structures composed of several single free forms roofed with separate individual transformed shells, are analyzed. The justification for creating roof structures composed of several shell segments results from the geometrical and mechanical properties of the transformed folded sheets employed. The shell folds are twisted around their longitudinal axes or twisted and bent transversely to these axes. That is the reason why the transverse ends of these folds expand and their middles contract as the degree of the transformation increases [13].

Therefore, it is not possible to combine two transformed corrugated shells along their transverse edges into one smooth shell (Figures 2 and 4) [32]. It is possible, however, to join both shells with transverse edges, so that there is an edge between them. The edge disturbs the smoothness of the resultant shell that becomes a structure of two shells. Most often, roof directrices separate the adjacent shell segments in the roof structure, or the adjacent shell segments are separated by additional roof or wall areas that let sunlight into the building's interior.

The main goals of combining complete transformed shells in the roof structure include: increasing the span of the roof and entire building, integrating the roof and façade forms, increasing the visual attractiveness of the entire building free form and making it sensitive to the natural or built environments. The concept most commonly used in the shaping of transformed folded shell structures is the combination of central sections of right hyperbolic paraboloids, their halves or quadrants in various configurations along their common edges (Figure 2a) [25,26].

The variety of shell structures constructed in this way is minimal. Reichhart's actions are also limited to structures composed of several identical central segments of right hyperbolic paraboloids additionally arranged on the same plane (Figure 2) [9].

The author presented wide possibilities of shaping free form structures composed of many individual free forms [13,17]. He developed the concept and coherent rules for creating such complex structures covered with plane-walled folded elevations and multi-segment transformed shell roof structures [18,19]. The developed algorithms allowed a radical increase in the variety of shapes of these forms [19,22,32,33]. Based on these algorithms, the author developed three methods presented in this work. These methods differ in quality and serve to obtain mutually different and specific goals. Each of these methods is aimed at creating building structures of very specific forms, in a convenient and relatively simple way. Therefore, in the author's opinion, only a qualitative comparison of these methods is justified.

The methods and examples presented in the sections that follow describe step by step and define objects, actions and algorithms used to solve increasingly complex issues of shaping internally consistent forms of building structures sensitive to the natural or built environments. The structure of the present article has been adopted so as to discuss step by step the specifics of the search for more and more sophisticated forms of building free forms roofed with transformed corrugated steel shell structures.

The designer may have to face, and cope with, some problems that arise from using unconventional methods for shaping general architectural forms of buildings roofed with transformed folded steel sheets and striving for relatively simple implementation of the designed innovative forms. The main task is to achieve geometrical, architectural and structural cohesion of all elements of each free form building, its shell roof in particular [15,32]. This aim can be accomplished by creating a parametric description of such building free forms and their specific structural systems based on the geometrical and mechanical characteristics of the transformed sheeting [14]. The proposed methods contain geometrical descriptions and algorithms that can be employed in the creating of parametric description of the free form structures covered with plane-walled folded elevations and complex transformed steel roof shell structures and writing parametric computer applications assisting the designer in the engineering developments.

Prokopska and the author continue the problems initiated by Reichhart. They propose a method of geometrical integration of each shell roof form with plane and oblique walls to obtain innovative, attractive and multi-variant architectural forms considered as morphological systems of buildings [18]. Some main principles of shaping complete and compound innovative free forms are the result of the cooperation between Prokopska and the author [32].

On the basis of these principles the author invented two methods for parametric shaping of the complete architectural free forms and their complex structures covered with transformed folded shell steel sheeting [18,22,31]. He assumed that the great freedom in shaping diversified transformed shell forms for roofing, resulting from great freedom in adopting shapes and mutual positions of roof directrices, can be used to integrate the entire building free form and make the form very sensitive to the natural or built environments [32]. Consequently, to achieve more consistent and sensitive architectural free forms, he decided to fold and incline elevation walls to the vertical depending on the shape of the shell roof and entire building. He noticed that the interdependence between the efficiency of the roof sheeting transformation and the location of its contraction along the length of each shell roof fold greatly enhances the attractiveness of the entire form and the integrity of the shapes of the roof and elevation [17].

Prokopska conducted multivariate interdisciplinary analyses of some consistent morphological systems that can be designed in harmony with the natural or man-made environments. Her research involves many interdisciplinary topics needed to develop experience in shaping various attractive architectural free forms [34,35]. Some of the proposed structural systems [36,37] can be modified and employed in the discussed building free forms [17].

### 3. Aims and Scope of the Article

The aim is to present new possibilities for the geometrical shaping of the free form structures of buildings roofed with many transformed shell segments, using three methods that differ in the complexity of algorithms, the purposes they can be used for, and regularities. The methods, presented in such a proposed order, are increasingly sophisticated in the creative search for coherent forms of the complex free buildings sensitive to the natural or built environments. They allow for obtaining these structures that differ in qualitative rather than quantitative properties of free forms, whose creation is discussed in the article.

Especially in the third method—the most complex and sophisticated method—a regular polyhedral network composed of many regular specific tetrahedrons is defined, such that the position of their side edges is optimized in relation to a finite number of selected straight lines normal to almost any double curved auxiliary regular surface, called the reference surface. The proposed rules, objects and activities ensure the regularity of the roof structure, integration of the structure with the folded façade form and allow the free form building to be adapted to the natural or built environments.

The dimensions of the roof shell and façade walls, and the inclination of their characteristic edges can be freely and creatively shaped as well as modified in the consequent steeps of the algorithms proposed by these methods, according to the expected engineering developments.

### 4. The Concept and the Range of the Article

The article proposes three methods for shaping complex free form buildings roofed with structures of corrugated shells made up of nominally flat folded sheets connected to each other by longitudinal edges, and transformed into spatial forms. The presentation of the methods on specific examples began with the simplest formulation based on a few very simple rules. The next two methods are increasingly complex and serve to achieve different sophisticated goals. Therefore, only a qualitative comparison of these three methods can be made. Comparison of the quantitative results achieved with the methods and included those in the specific examples presented is not justified. The quantitative comparison of these methods with the methods of other authors mentioned in the previous sections also seems unjustified.

The first method formulates the basic conditions that must be met by single free forms $\Sigma$ covered with single transformed shells $\Omega$ (Figure 6), so that they can be combined into a structure in a simple way. The combined structure has a free form, and is roofed with a structure composed of several shell segments. The purpose of creating a roof structure is to increase the span of a complex building form in relation to the span of a single form.

The basic action of the first method is to create a model of roof eaves of each complete free form. The model is a closed spatial quadrangle $B_{ev}$, whose geometrical properties depend on the form of rectangular, nominally flat sheets folded in one direction and transformed into a shell form. Points $B_i$ are four vertices of $B_{ev}$.

Since the folded sheets are rectangular, the angles between two adjacent sides having shared ends at corners $B_j$ ($j = 1$–$4$) of the spatial quadrangle $B_{ev}$ are very close to right angles, and the lengths of each pair of the opposite sides of the quadrangle are equal to each other or differ very little. Two opposite sides of the aforementioned quadrangle, corresponding to the transverse edges of a transformed shell are almost equal to each other. Furthermore, if the transverse edges of the shell are not obliquely cut [13], the lengths of all folds of the shell are identical, so the second pair of the opposite sides of this quadrangle is formed from two skew straight sections of equal length. The transverse fold ends are often cut obliquely to adjust the transverse edges of the shell to the direction of the roof directrices. However, the cuts are only minor and cause little variation in the fold's length, followed by a slight difference in the lengths of the opposite sides and the measures of the corner angles of the aforementioned spatial quadrangle.

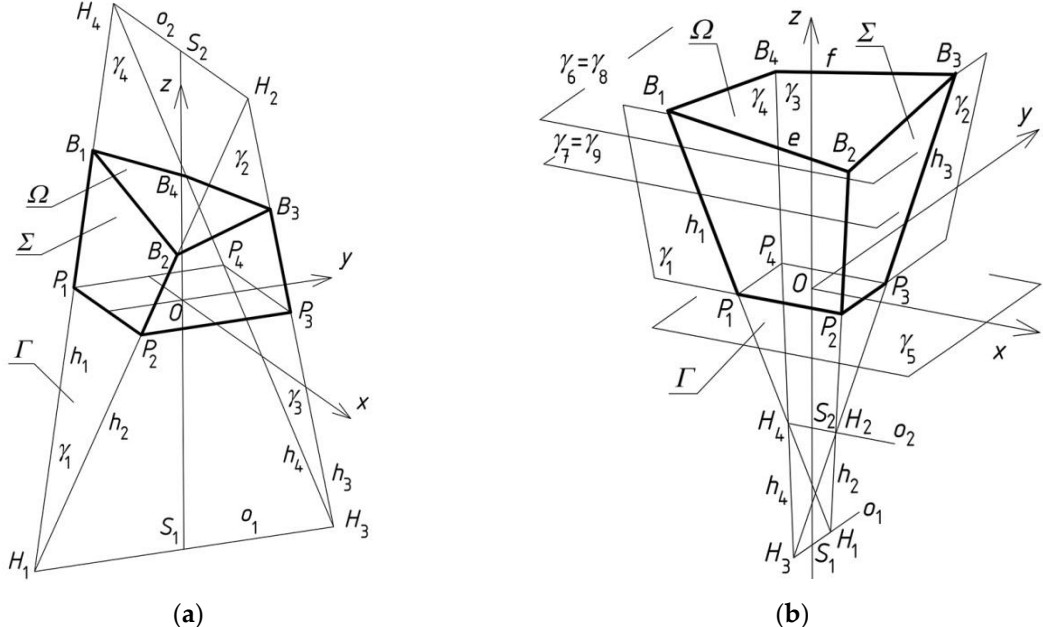

**Figure 6.** Two simplified models $\Sigma$ of a free form building roofed with transformed shells $\Omega$: (**a**) straight directrices; (**b**) curved directrices $e$ and $f$.

As all folds of each transformed roof shell are almost always twisted along the longitudinal axes, expanded at the transverse ends and contracted at half-length, each quadrangle cannot be flat; on the contrary, it must be spatial. Each pair of its two opposite sides is created as two skewed lines. The lengths of these sides must be precisely calculated based on the border conditions adopted for the roof's corrugated shell. The conditions are determined mainly by the shape and mutual position of the roof directrices [14,18].

For the above models and in the initial example presented in Section 5, the directrices are adopted as two straight segments $e = B_1 B_2$ and $f = B_3 B_4$ in order to obtain a relatively simple description of the first method. The $B_{ev}$ spatial quadrilateral is made up of two pairs of skew straight sections. One pair of these sections is formed of the $e$ and $f$ directrices corresponding to the shell's edges running transversely to the shell fold directions. The second pair of the opposite straight sections corresponds to the longitudinal edges of the shell belonging to the extreme folds.

Since the directrices are skewed straight lines, the calculated supporting conditions of the subsequent folds in a shell, mainly affecting the fold's twist, are varied. As a result, the subsequent folds have different lengths of their supporting lines, twists and lengths. The author attempts to use quadrangles $B_{ev}$ whose shapes are symmetrical towards an axis skew with respect to each side of quadrangle $B_{ev}$ in order to obtain pairs of its opposite edges of equal length and congruent apex angles. Such an operation leads to a symmetrical complete roof shell, identical inclination of each shell fold to both directrices and identical supporting conditions of the fold at both ends.

In order to build a very simple free form whose roof eaves are spatial quadrangle $B_{ev}$ characterized by the aforementioned properties, the definition of four planes $\gamma_i$ modeling four façade walls of this form is most convenient (Figure 6). In these planes, four straight sections of $B_{ev}$ with common vertices are defined. On the basis of these elements, the author defined a simplified model of a single free form, and called it the reference tetrahedron $\Gamma$ [13]. He distinguished three basic types of the reference tetrahedrons. Two of them are presented in Figure 6.

He also drew attention to the following basic geometrical properties of each reference tetrahedron $\Gamma_i$ of the free form structure $\Gamma$, where $i$ indicates the number of all reference tetrahedrons used for creating free form structure $\Sigma$. For each reference tetrahedron, two of the six edges formed as a result of the intersection of the above four planes are called axes $o_1$ and $o_2$ (Figure 6), and the other are side

edges $h_j$ ($j$ = 1–4). The axes are the intersecting lines of two opposite planes of the tetrahedron. The side edges are the intersection of the adjacent planes of tetrahedron $\Gamma_i$. The axes intersect with the side edges at points $H_j$ called the vertices of $\Gamma_i$. Vertices $B_j$ of each quadrangle $B_{evi}$ are defined at side edges $h_j$ at adequate distances from the respective vertices $H_j$. Four sides of $B_{ev}$ are created on the basis of four vertices $B_j$.

In order to obtain spatial network $\Gamma$ composed of many complete reference tetrahedrons $\Gamma_i$, the reference tetrahedrons are arranged so that one wall of each two neighboring tetrahedrons is contained in a common plane. A detailed description of the relations between the axes, edges and vertices of neighboring reference tetrahedrons $\Gamma_i$ in $\Gamma$ is presented in Section 5 concerning the first method proposed by the author.

The method enables the full recognition of the possibilities of combining the reference tetrahedrons into one spatial reference network $\Gamma$ whose unconventional and innovative form is determined by the specific properties of these tetrahedrons $\Gamma_i$. The method leaves the designer complete freedom to give the shape of the spatial reference network $\Gamma$ and the roof shell structure $\Sigma$. As a result, the network may be regular or not. The arrangement of the shell segments in the three-dimensional space may be regular or not. The forms of the individual shells may be regular and similar to each other or not, depending on the geometrical properties of the subsequently adopted complete free forms.

As reference network $\Gamma$ is used to create regular free form building structures roofed with regular shell roof structures, it is necessary to define additional rules supporting the designer's spatial reasoning and intuition. Therefore, the author developed the second method of shaping the aforementioned structures roofed with transformed curved shells. In order to impose the regularity of a complex free form and its roof shell structure, a so-called reference surface is introduced into the method. According to the method, the shapes and arrangement of the reference tetrahedrons in the three-dimensional space should be determined on the basis of the reference surface.

Using this method, it is possible to exploit specific geometrical properties of the reference surface, such as its planes of symmetry, so that the symmetry axes of the largest possible number of reference tetrahedrons $\Gamma_i$ are contained in these planes. Moreover, relatively simple operations are possible to obtain the symmetry of quadrangles $B_{evi}$ contained in $\Gamma_i$. In this method, however, reference tetrahedrons $\Gamma_i$ are arranged on the basis of the reference surface in an intuitive manner to a large extent, whereby their construction is related to the curvatures of the reference surface only to a small extent. This may lead to unjustified differences in the shapes and irregular distribution of the reference tetrahedrons in relation to the reference surface, so certain additional rules are needed. Therefore, the author decided to employ some planes normal to the reference surface for selected planes of the reference tetrahedrons.

In order to prevent inefficient and irregular forms of the reference network and shell roof structure, the author created the third method and introduced another condition allowing him to optimize the regular shapes and arrangement of the reference tetrahedrons in relation to the arbitrary reference surface. He imposed a constraint that the directions of all side edges of the reference network should be close to the directions of the adopted normals of the reference surface at the points of the intersection of the above side edges and the reference surface within the adopted optimization accuracy. Such operations, undertaken by the author, result in the fact that the projections of the single roof shell segments on a reference surface, in the directions compatible with the directions of selected straight lines normal to this surface, do not overlap each other. Instead, they form a continuous two-dimensional surface. In other words, the projections are not disjoint and do not create discontinuous areas.

The third method allows an optimization of the positions and directions of the side edges $h_{i,j}$ of reference tetrahedrons $\Gamma_{i,j}$ relative to the arbitrary reference surface. The optimization is carried out using the aforementioned straight lines $n_{i,j}$ normal to the reference surface. The optimizing condition is the allowable size of the deviation of all edge sides from the corresponding normals to the reference surface.

The basic difficulty in maintaining the above optimizing condition is the fact that two subsequent straight lines $n_{i,j}$ normal to the reference surface are skew lines, in contrast to each pair of neighboring edges $h_{i,j}$ of each reference tetrahedron $\Gamma_{i,j}$, which intersect each other at the appropriate vertex $H_{i,j}$ of $\Gamma_{i,j}$. Therefore the perfect approximation or replacement of $n_{i,j}$ with $h_{i,j}$ is impossible, and the optimization of the side edges $h_{i,j}$ in relation to the reference surface is needed. In the third method, the author therefore included an innovative way of looking for a regular spatial network composed of regular reference tetrahedrons whose side edges $h_{i,j}$, intersecting each other at vertices $H_{i,j}$ of reference tetrahedrons $\Gamma_{i,j}$, are optimized and defined on the basis of not crossing each other's normals with regard to the reference surface.

## 5. Structures as Compositions of Many Regular Free Forms

The following steps, activities and objects have been identified in the algorithm of the first developed method for shaping complex free form structures. In the first step, some actions are undertaken to build the general form of a single free form with the help of reference tetrahedron $\Gamma_1$, whose four walls model four façade walls of the designed free form.

In the second step, the roof of a single free form $\Sigma_1$ is modeled as a sector of a smooth regular warped surface. The roof is made of nominally flat corrugated sheets transformed effectively into a shell shape; that is, the shell shape resulting from this transformation should contract at half-length of each fold, transversally to the fold directions, which positively affects the static-strength work of the folds in the shell.

In the third step of the algorithm, a method for determining the positions of several individual forms and combining these forms into a structure sensitive to the predicted natural or man-made environments is carried out. The complex building free form is, therefore, the sum of several individual free forms appropriately set together with the common façade walls. The roof of the building structure created this way is a shell roof structure composed of several smooth shell segments.

In the last step of the algorithm, modification of the forms of the roof and façade of the previously achieved structure is possible. This modification is based on displacements of selected roof edges or façades, in the planes of the reference tetrahedrons employed. The purpose of this modification is to make the building structure more sensitive to the built or natural environments. Complete reference tetrahedron $\Gamma_1$ (Figure 7a) is formed by means of four vertices: $H_{1,1}$, $H_{2,1}$, $H_{3,1}$, $H_{4,1}$. To determine the positions of the above four vertices, two skewed straight lines $o_{1,1}$ and $o_{2,1}$ located in distance $d_{n1}$ and perpendicular to each other are assumed. Middle point $S_{2,1}$ of $H_{2,1}H_{4,1}$, is lain in the distance $d_{p1}$ from the origin $O$ of the orthogonal coordinate system $[x,y,z]$. A straight line $z$ perpendicular to $o_{1,1}$ and $o_{2,1}$ intersects these lines at points $S_{1,1}$, and $S_{2,1}$. Finally, the vertices $H_{3,1}$, $H_{1,1}$ are measured along $o_{t1,1}$ in the distances $d_{3,1}$ and $d_{1,1}$ from $S_{1,1}$, and the vertices $H_{4,1}$, $H_{2,1}$ are measured along $o_{2,1}$ in the distances $d_{4,1}$ and $d_{2,1}$ from $S_{2,1}$.

Lines $o_{1,1}$, and $o_{2,1}$ are called the axes of $\Gamma_1$. However, straight lines: $h_{1,1}(H_{1,1}, H_{4,1})$, $h_{2,1}(H_{1,1}, H_{2,1})$, $h_{3,1}(H_{2,1}, H_{3,1})$, $h_{4,1}(H_{3,1}, H_{4,1})$ are said to be the side edges of $\Gamma_1$. In order to obtain the vertices: $P_{1,1}$, $P_{2,1}$, $P_{3,1}$, $P_{4,1}$ of a planar base of $\Gamma_{1,1}$, plane $(x, y) \perp z$ is passed through point $O$ located in the distance $d_{p1}$ from $S_{2,1}$, (Figure 7a). Vertices $B_{1,1}$, $B_{2,1}$, $B_{3,1}$, $B_{4,1}$ of the free form $\Sigma_1$ are constructed on $h_{1,1}$, $h_{2,1}$, $h_{3,1}$, $h_{4,1}$ in the distances $d_{h1,1}$, $d_{h2,1}$, $d_{h3,1}$, $d_{h4,1}$ from the above points $P_{i,1}$ ($i = 1$–4). Values of all above input data are presented in Table 1. The above four values are adopted so that the eaves of the resultant $\Sigma_1$ are characterized by two pairs of opposite segments of equal length to obtain a central sector of an oblique hyperbolic paraboloid [19].

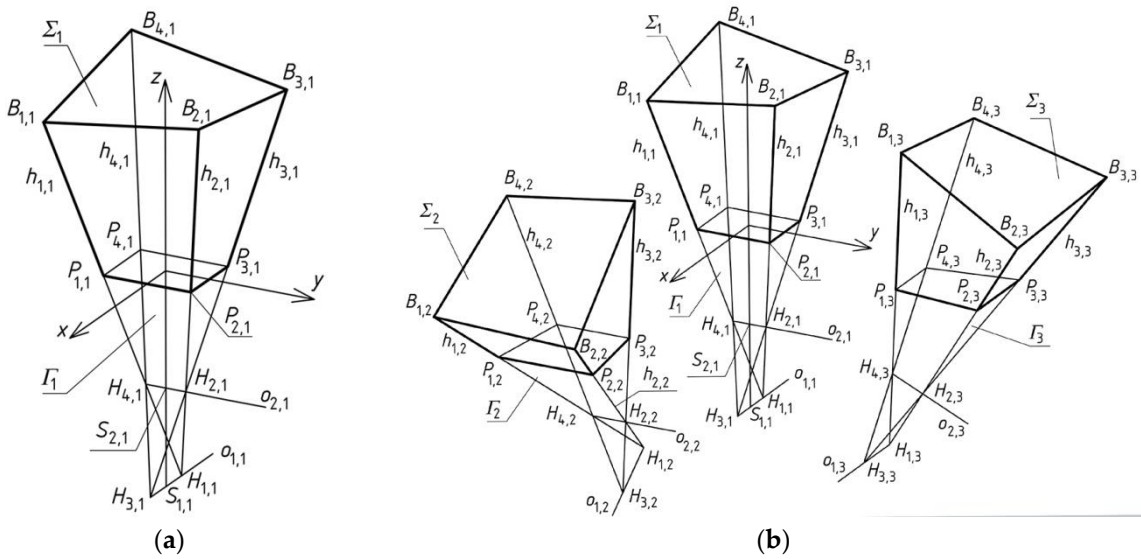

**Figure 7.** (**a**) Properties of a reference tetrahedron; (**b**) Composition of three reference tetrahedrons located orthogonally in reference structure.

**Table 1.** Parameters providing the parametric characteristics of reference tetrahedron $\Gamma_1$ and free form structure $\Sigma_1$.

| Parameter | Value |
|:---:|:---:|
| $d_{1,1} = d_{3,1}$ | 3310.6 |
| $d_{2,1} = d_{4,1}$ | 2231.8 |
| $d_{n1}$ | 10,500.0 |
| $d_{p1}$ | 12,500.0 |
| $d_{h2,1}$ | 21,827.3 |
| $d_{h3,1}$ | 18,827.3 |
| $d_{h1,1}$ | 18,827.3 |
| $d_{h4,1}$ | 21,827.3 |

To determine the subsequent reference tetrahedrons of reference tetrahedral structure $\Gamma$ being sought, one should create nine complete free forms $\Gamma_j$ ($j = 1$–9) of $\Gamma$. In order to create reference tetrahedron $\Gamma_2$, four vertices $H_{i,2}$ ($i = 1$–4) (Figure 7b) should be adopted as previously the vertices of $\Gamma_2$. In order to create free form $\Sigma_2$, vertices $B_{i,1}$ and $P_{i,1}$ ($i = 1$–4) should be adopted on the basis of the appropriate parameters as previously.

The same action must be performed to obtain reference tetrahedron $\Gamma_3$ and free form $\Sigma_3$ (Figure 7b). However, for the designed reference polyhedral structure $\Gamma$, other activities are undertaken to simplify the assembly of reference tetrahedrons $\Gamma_j$ ($j = 1$–9) into one structure $\Gamma$ and complete free forms $\Sigma_j$ ($j = 1$–9) into one structure $\Sigma$.

According to the aforementioned concept, it is taken that $H_{3,2} = H_{1,1}$, $H_{2,2} \in h_{1,2}$, $H_{4,2} \in h_{1,1}$ (Figure 8). It is assumed that $H_{2,2} = H_{2,1}$, $H_{4,2} = H_{4,1}$. To obtain $\Gamma_2$ symmetrical to $\Gamma_1$ towards plane $\xi_2$ ($H_{3,2}$, $H_{2,2}$, $H_{4,2}$), vertex $H_{1,2}$ being sought has to be symmetrical to $H_{3,1}$ towards $\xi_2$.

The transformation related to plane $\xi_2$ of symmetry is denoted as $L_2$, so $H_{1,2} = L_2(H_{3,1})$, $B_{1,2} = L_2(B_{4,1})$, $B_{2,2} = L_2(B_{3,1})$, $\Gamma_2 = L_2(\Gamma_1)$. In addition, $B_{3,2} = B_{2,1}$, $B_{4,2} = B_{1,1}$, $P_{3,2} = P_{2,1}$, $P_{4,2} = P_{1,1}$ and $P_{1,2} = (H_{1,2}, H_{4,2}) \cap (x, y)$ and $P_{2,2} = (H_{1,2}, H_{2,2}) \cap (x, y)$.

Reference tetrahedron $\Gamma_3$ is created in the same way as $\Gamma_2$ so that plane $\xi_3$ ($H_{4,3} = H_{2,1}$, $H_{3,3} = H_{3,1}$, $H_{1,3} = H_{1,1}$) is used for transformation $L_3$. Thus, $B_{3,3} = L_3(B_{4,1})$, $B_{2,3} = L_3(B_{1,1})$, $\Gamma_3 = L_3(\Gamma_1)$.

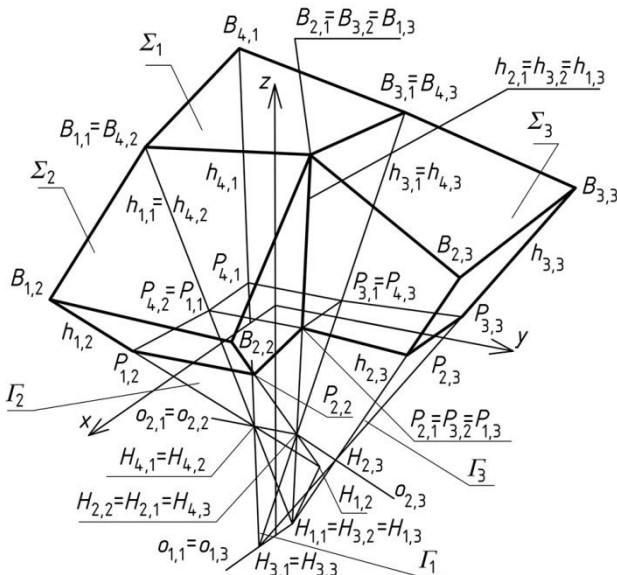

**Figure 8.** Three reference tetrahedrons: $\Gamma_1$, $\Gamma_2$ and $\Gamma_3$ located orthogonally in the polyhedral reference structure.

$\Gamma_2$ and $\Gamma_3$ are located orthogonally in $\Gamma$. The reference tetrahedron $\Gamma_4$ being sought is located diagonally in $\Gamma$. The way of creating $\Gamma_4$ is different from $\Gamma_2$ and $\Gamma_3$ because the locations of two its vertices are known: $H_{4,4} = H_{2,1}$, $H_{3,4} = H_{1,1}$ (Figure 9). The searched vertices $H_{1,4}$ and $H_{2,4}$ have to belong to $h_{1,4} = h_{2,2}$ and $h_{3,4} = h_{2,3}$, respectively.

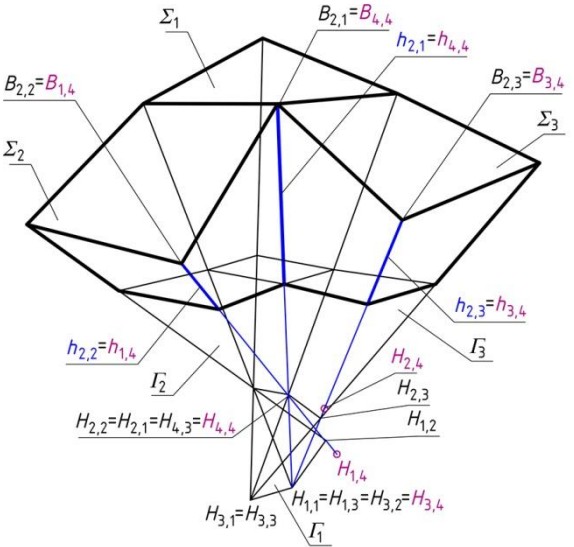

**Figure 9.** Construction of the forth diagonal reference tetrahedron $\Gamma_4$ on the basis of three reference tetrahedrons: $\Gamma_1$, $\Gamma_2$ and $\Gamma_3$ located orthogonally in the polyhedral reference structure.

To obtain $\Sigma_4$, vertices $H_{1,4}$ and $H_{2,4}$ are accepted at side edges $h_{1,4}$ and $h_{3,4}$ (Figure 10). It is assumed that $B_{4,4} = B_{2,1}$, $B_{3,4} = B_{2,3}$, $B_{1,4} = B_{2,2}$. The position of $B_{2,4}$ is determined at $h_{2,4}(H_{1,4}, H_{2,4})$. Point $P_{2,4}$ is the intersection of $h_{2,4}$ and the base plane $(x,y)$. The data used in the present example are given in Tables 2 and 3.

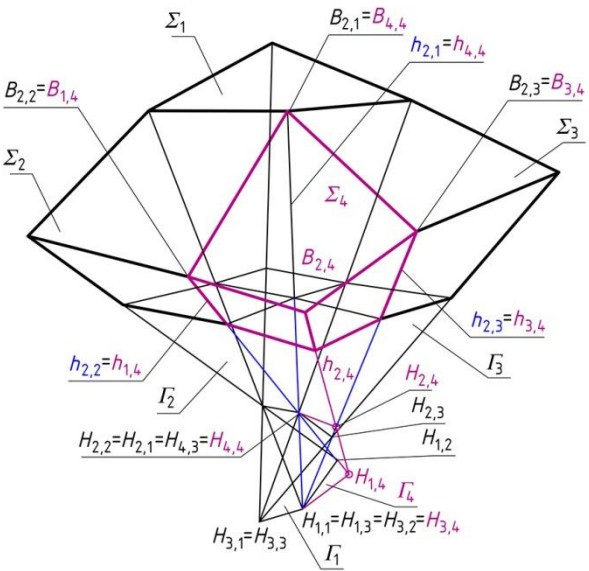

**Figure 10.** Construction of the forth diagonal reference tetrahedron on the basis of three reference tetrahedrons: $\Gamma_1$, $\Gamma_2$ and $\Gamma_3$ located orthogonally in the polyhedral reference structure.

**Table 2.** Coordinates of the roof and elevations edges vertices of basic free form $\Sigma$.

| Vertex | X-Coordinate | Y-Coordinate | Z-Coordinate |
|--------|-------------|-------------|-------------|
| $B_{2,1}$ | 10,373.9 | 9225.2 | 20,402.1 |
| $B_{3,1}$ | −9489.7 | 8629.2 | 17,598.0 |
| $B_{1,1}$ | 9489.7 | −8629.2 | 17,598.0 |
| $B_{4,1}$ | −10,373.9 | −9225.2 | 20,402.1 |
| $B_{2,2}$ | 25,036.9 | 8629.2 | 6711.9 |
| $B_{1,2}$ | 27,369.5 | −9225.2 | 8501.8 |
| $B_{2,3}$ | 9489.7 | 24,395.6 | 10,578.5 |
| $B_{3,3}$ | −10,373.9 | 26,080.6 | 12,897.8 |
| $B_{2,4}$ | 29,067.4 | 26,269.0 | 6580.0 |
| $P_{2,1}$ | 3941.2 | 4888.7 | 0.0 |
| $P_{3,1}$ | −3941.2 | 4888.7 | 0.0 |
| $P_{1,1}$ | 3941.2 | −4888.7 | 0.0 |
| $P_{4,1}$ | −3941.2 | −4888.7 | 0.0 |
| $P_{2,2}$ | 16,290.0 | 6394.2 | 0.0 |
| $P_{1,2}$ | 16,290.0 | −6394.2 | 0.0 |
| $P_{2,3}$ | 5457.2 | 16,710.1 | 0.0 |
| $P_{3,3}$ | −5457.2 | 16,710.1 | 0.0 |
| $P_{2,4}$ | 19,496.9 | 19,889.3 | 0.0 |

**Table 3.** Coordinates of the vertices of reference structure $\Gamma$.

| Vertex | X-Coordinate | Y-Coordinate | Z-Coordinate |
|--------|-------------|-------------|-------------|
| $H_{1,1}$ | −3310.6 | 0.0 | −2300.0 |
| $H_{3,1}$ | 3310.6 | 0.0 | −2300.0 |
| $H_{2,1}$ | 0.0 | 2231.8 | −12,500.0 |
| $H_{4,1}$ | 0.0 | −2231.8 | −12,500.0 |
| $H_{1,2}$ | −8734.4 | 0.0 | −19,202.3 |
| $H_{2,3}$ | 0.0 | 6309.5 | −14,315.5 |
| $H_{1,4}$ | −11,337.6 | −665.2 | −21,199.8 |
| $H_{2,4}$ | 470.4 | 7206.1 | −13,081.4 |

In addition, the following dependences should be adopted: $B_{1,3} = B_{2,1}$, $B_{4,3} = B_{3,1}$, $P_{1,3} = P_{2,1}$, $P_{4,3} = P_{3,1}$, $B_{1,4} = B_{2,2}$, $B_{3,4} = B_{2,3}$, $B_{4,4} = B_{2,1}$, $P_{1,4} = P_{2,2}$, $P_{3,4} = P_{2,3}$, $P_{4,4} = P_{2,1}$, $H_{1,3} = H_{1,1}$, $H_{3,3} = H_{3,1}$,

$H_{4,3} = H_{2,1}$, $H_{3,4} = H_{3,2}$, $H_{4,4} = H_{2,1}$. Other vertices are symmetrical towards plane $(x,z)$ or $(y,z)$. Thus, in order to obtain the entire reference structure $\Gamma$ and free form structure $\Sigma$, four reference polyhedrons $\Gamma_i$ $(i = 1$–$4)$ and four free forms $\Sigma_i$ have to be transformed towards these planes of symmetry $(x,z)$ and $(y,z)$ into new positions of $\Gamma_r$ and $\Sigma_r$ $(r = 5$–$9)$. The final reference structure $\Gamma$ and free form structure $\Sigma$, are shown in Figure 11.

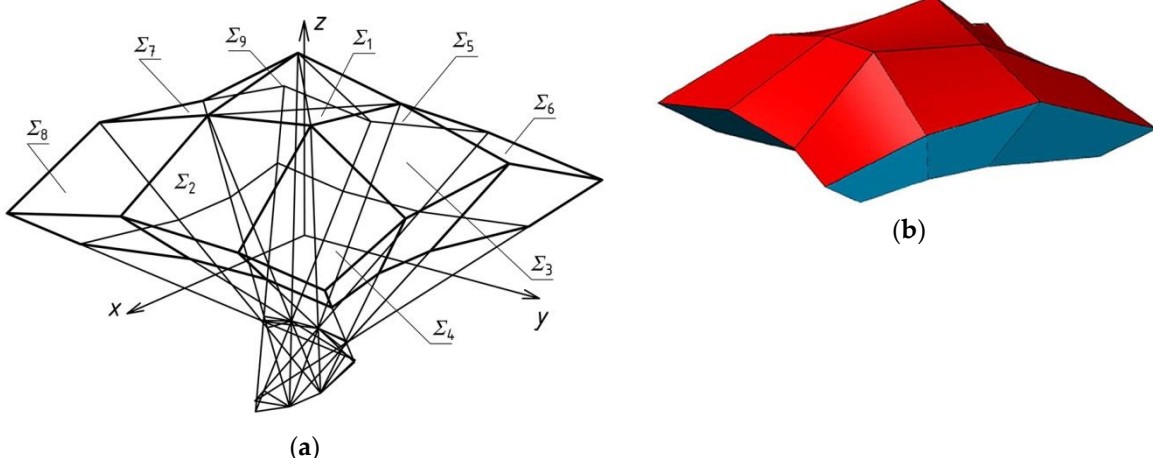

**Figure 11.** The free form structure created on the basis of the polyhedral reference structure: (**a**) an edge model; (**b**) a shell model.

It is worth stressing that the reference tetrahedrons located orthogonally or diagonally in $\Gamma$ do not have to be congruent to each other, so the form of the final $\Gamma$ and $\Sigma$ can take unsymmetrical shapes. Thus, the possible shapes of $\Sigma$ may be really free, diversified and sensitive to the built and natural environments.

The free form presented previously is the basis of creating some derivative forms. The derivative free forms are shaped as a result of displacing or rotating some selected side edges of the basic free form in selected planes of its reference structure. The basic form is covered with the continuous shell structure (Figure 11b), whose individual shells are divided by shared edges locally disturbing the smoothness of the structure and forming a regular pattern on the roof. In addition, the elevations have relatively simple shapes.

The first derivative form is constructed as a result of displacing the selected vertices belonging to five roof border quadrangles located diagonally in $\Gamma$ and distinguished by means of a black thick line in Figure 12. Their selected vertices $B_{i,j}$ are displaced along the relevant side edges $h_{i,j}$ in the distances equal to $d_h = 3000.0$ mm. The values of the coordinates of the points used are presented in Table 4.

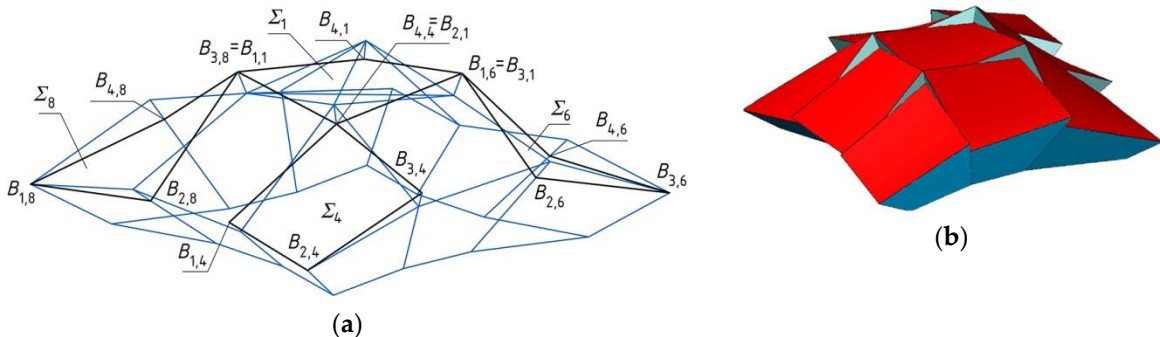

**Figure 12.** The new transformed discontinuous free form structure created on the basis of the continuous free form structure presented earlier: (**a**) an edge model; (**b**) a shell model.

**Table 4.** Coordinates of the selected roof vertices of the first derivative form.

| Vertex | X-Coordinate | Y-Coordinate | Z-Coordinate |
|---|---|---|---|
| $B_{2,1}$ | 9489.7 | 8629.2 | 17,598.0 |
| $B_{1,1}$ | 10,373.9 | −9225.2 | 20,402.1 |
| $B_{1,4}$ | 27,369.5 | 9225.2 | 8501.8 |
| $B_{3,4}$ | 10,373.9 | 26,080.6 | 12,897.8 |

For the aforementioned free form, the parametric shape characteristics can be improved and extended by new shape parameters, for example describing the proportion of the roof discontinuity areas intended for windows to the area of the entire shell roof. Propositions of such additional shape parameters go beyond the scope of the paper.

The second derivative form (Figure 13) is created as a result of: (a) the translations of all vertices of the eaves of $\Sigma_1$ discussed earlier, along the relevant side edges of $\Gamma_1$ in equal distances $d_h$; and (b) the translations of the selected elevation side edges along the selected axes of the reference structure in the distances $d_{o1,2} = H_{3,2}H_{1,1} = 4534.1$ mm and $d_{o2,1} = H_{2,3}H_{4,3} = 3126.7$ mm, so that these side edges will be contained in the planes of control compositions $\Gamma_2$ and $\Gamma_3$. The values of the parameters used are as follows: $B_{3,1}B_{4,4} = B_{3,1}B_{2,1} = d_{h1,1} = 3000.0$ mm. The values of the coordinates of the transformed vertices of the eaves are included in Table 5.

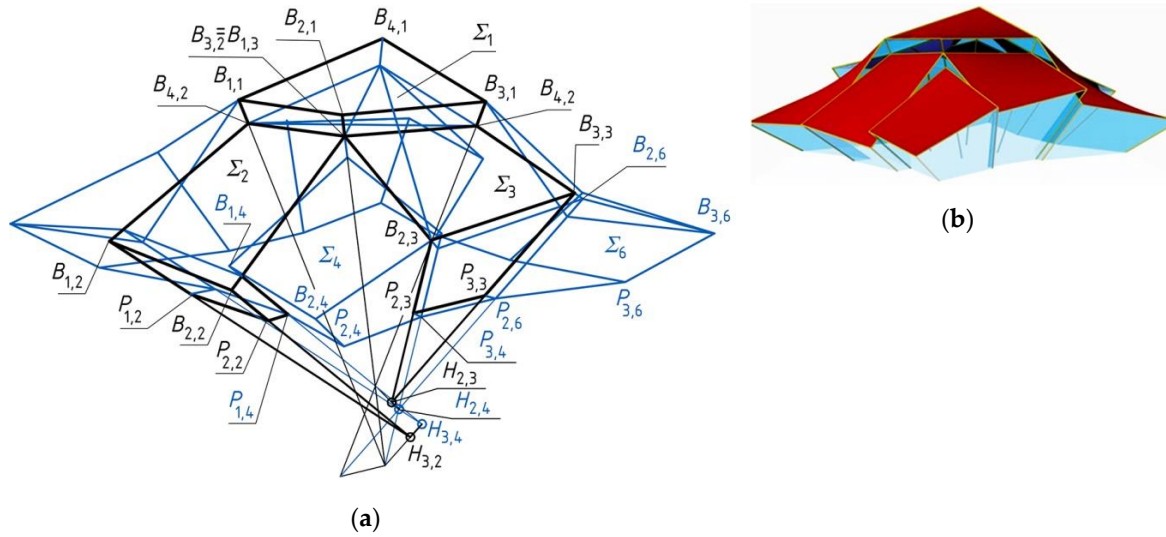

(**a**)

(**b**)

**Figure 13.** The transformed discontinuous free form structure created on the basis of the continuous free form structure presented earlier: (**a**) an edge model; (**b**) a shell model.

**Table 5.** Coordinates of the roof and elevations vertices of the second derivative form.

| Vertex | X-Coordinate | Y-Coordinate | Z-Coordinate |
|---|---|---|---|
| $B_{2,1}$ | 11,258.0 | 9821.2 | 23,206.2 |
| $B_{1,1}$ | 10,373.9 | −9225.2 | 20,402.1 |
| $B_{2,2}$ | 26,511.9 | 8569.2 | 5334.8 |
| $B_{1,2}$ | 29,335.6 | −9290.8 | 7501.5 |
| $B_{2,3}$ | 9553.8 | 23296.3 | 11,290.3 |
| $B_{3,3}$ | −10,305.4 | 24728.7 | 13,261.9 |
| $P_{2,2}$ | 19,559.6 | 6792.8 | 0.0 |
| $P_{1,2}$ | 19,559.6 | −6792.8 | 0.0 |
| $P_{2,3}$ | 5249.9 | 15,093.6 | 0.0 |
| $P_{3,3}$ | −5249.9 | 15,093.6 | 0.0 |

## 6. Structures Based on Regular Spatial Networks

A new method for shaping the free form structures is presented in the example of nine complete shells located towards a reference ellipsoid $\omega$ (Figure 14) is described. Ellipsoid $\omega$ is expressed as:

$$\frac{x^2}{a^2} + \frac{y^2}{b^2} + \frac{z^2}{c^2} = 1 \tag{1}$$

where $a$ = 25,000 mm, $b$ = 20,000 mm, $c$ = 13,000 mm.

The orthogonal coordinate system [$x$, $y$, $z$] having its origin at centre $O$ of $\omega$ (Figure 14) is adopted. Three basic ellipses $w_0$, $t_0$, $u_0$ are the intersection of $\omega$ with planes ($y$, $z$), ($x$, $z$) and ($x$, $y$), respectively. In the examples presented below, the symbols of all created objects, for example reference tetrahedrons and their vertices, have been changed for a more consistent description of creating reference networks.

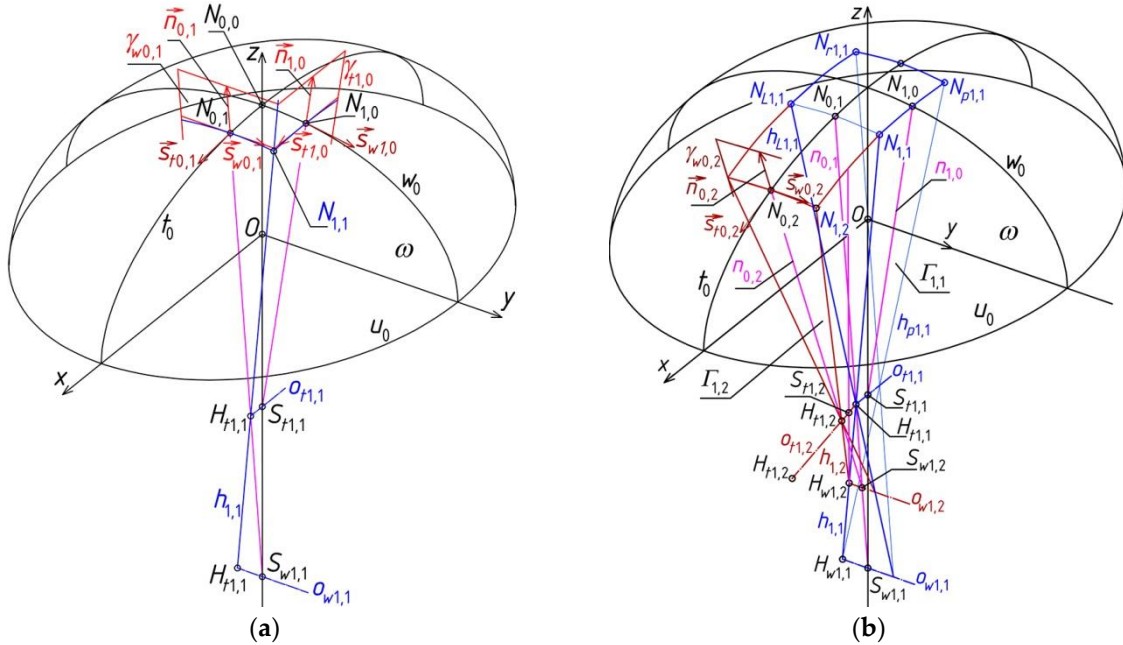

(a)　　　　　　　　　　　　　　　　　　　(b)

**Figure 14.** Shaping of: (**a**) a complete free form with the help of a reference tetrahedron; (**b**) a free form structure with the help of few reference tetrahedrons.

The planes $\gamma_{w0,1}$ and $\gamma_{t1,0}$ from among four planes of the first reference polyhedron $\Gamma_{1,1}$ are presented in Figure 14. The positions of these four planes of $\Gamma_{1,1}$ are obtained so that these planes are normal to $\omega$ and $w_0$ or $t_0$. Points $N_{0,0}$, $N_{0,1}$ and $N_{1,0}$ have to be found to obtain $\Gamma_{1,1}$. $N_{0,0} = z \cap \omega$, $N_{0,1} \in t_0$, and $N_{0,1}$ is located it the distance $d_{N01}$ = 5000 mm from $N_{0,0}$, and $N_{1,0} \in w_0$ as well as $N_{1,0}$ is lain in the distance $d_{N10}$ = 4500 mm from $N_{0,0}$. The planes $\gamma_{w0,1}$ and $\gamma_{t1,0}$ pass through $N_{0,1}$ or $N_{1,0}$ perpendicularly to $t_0$ or to $w_0$. The unit vectors $n_{0,1} = [l_{n0,1}, m_{n0,1}, n_{n0,1}]$ and $n_{1,0} = [l_{n1,0}, m_{n1,0}, n_{n1,0}]$ are normal to ellipsoid $\omega$. They determine, with the unit vectors $s_{t0,1}$ or $s_{w1,0}$ tangent to $t_0$ or $w_0$ at $N_{0,1}$ and $N_{1,0}$, two planes $\gamma_{w0,1}$ and $\gamma_{t1,0}$ of $\Gamma_{1,1}$. Thus, vectors $n_{1,1}$, $n_{0,1}$ and $n_{1,0}$ normal to $\omega$ at $N_{1,1}$, $N_{0,1}$ and $N_{1,0}$ are helpful in determining $\Gamma_{1,1}$. In order to determine vector $n_{0,1}$, vector $s_{w0,1}$ $||$ $y$ can be passed through $N_{0,1}$, so $n_{0,1} = s_{w0,1} \times s_{t0,1}$ (Figure 14a). By analogy, $n_{1,0} = s_{w1,0} \times s_{t1,0}$. $N_{1,1}$ is created as a result of the intersection of $\omega$ with edge $h_{1,1} = \gamma_{w0,1} \cap \gamma_{t1,0}$. $\Gamma_{1,1}$ is symmetrical towards ($x$, $z$) or ($y$, $z$).

Side edge $h_{1,1}$ is not identical with straight line $n_{1,1}$ normal to $\omega$ at $N_{1,1}$ but only close to that line. An action leading to such a situation that the direction of $h_{1,1}$ is the closest possible to the direction of $n_{1,1}$ is expected. It may be obtained by changing the inclination of $\gamma_{w0,1}$ to ($y$, $z$) and the inclination of $\gamma_{t1,0}$ to ($x$, $z$), so the inclination of $h_{1,1}$ towards $n_{1,1}$ is also changed. The control of the above changes

so that the angle of the inclination of $h_{1,1}$ to $n_{1,1}$ will be equal to the angles between the new and old positions of $\gamma_{w0,1}$ and $\gamma_{t1,0}$ is needed, however, this activity goes beyond the scope of this paper.

Reference tetrahedron $\Gamma_{1,1}$ is used for the central control composition of final reference structure $\Gamma$ composed of nine reference tetrahedrons $\Gamma_{i,j}$. The plane $\gamma_{w0,1}$ of $\Gamma_{1,1}$ is accepted as one of four planes of the new reference tetrahedron $\Gamma_{1,2}$. The other three planes of $\Gamma_{1,2}$ are constructed in the following order (Figure 14b): (a) point $N_{0,2} \in t_0$ on ellipse $t_0$ determined in the distance $d_{w0,2} = 5000$ mm from $N_{0,1}$; (b) plane $\gamma_{w0,2}$ passing through $N_{0,2}$ and normal to $t_0$; (c) straight line $o_{w1,2} = \gamma_{w0,2} \cap \gamma_{w0,1}$; (d) straight line $o_{t1,2}$ passing through point $H_{t1,1} = h_{1,1} \cap h_{L1,1}$ and parallel to $(N_{0,1}, N_{0,2})$. On the basis of the above elements and activities, the following sets are obtained: a) the tetrad of planes: $\gamma_{w0,1}$, $\gamma_{w0,2}$, $\gamma_{t1,2}(o_{t1,2}, h_{1,1})$ and $\gamma_{Lt1,2}$ symmetrical to $\gamma_{t1,2}$ towards $(x, z)$; and b) the tetrad of side edges of $\Gamma_{1,2}$: $h_{1,1}$, $h_{L1,1}$, $h_{1,2} = \gamma_{t1,2} \cap \gamma_{w0,2}$ and $h_{L1,2}$ symmetrical to $h_{1,2}$ towards $(x, z)$.

Reference tetrahedron $\Gamma_{2,1}$ is created in an analogous way as for $\Gamma_{1,2}$. Plane $\gamma_{t1,0}$ of $\Gamma_{1,1}$ is accepted as one of four planes of the new $\Gamma_{2,1}$ (Figure 15). The other three planes of $\Gamma_{2,1}$ are constructed by means of: a) point $N_{2,0} \in w_0$ constructed in the distance $d_{t2,0} = 4500$ mm from $N_{1,0}$; b) plane $\gamma_{t2,0}$ normal to $w_0$ and passing through $N_{2,0}$; c) straight line $o_{t2,1} = \gamma_{t1,0} \cap \gamma_{t2,0}$; and d) straight line $o_{w2,1}$ parallel to $(N_{1,0}, N_{2,0})$ and passing through point $H_{w1,1} = h_{1,1} \cap h_{p1,1}$. The following elements of $\Gamma_{2,1}$ are obtained: a) the tetrad of planes: $\gamma_{t1,0}$, $\gamma_{t2,0}$, $\gamma_{w2,1}$ $(o_{w2,1}, h_{1,1})$ and $\gamma_{pw2,1}$ symmetrical to $\gamma_{w2,1}$ towards $(y, z)$; and b) the tetrad of side edges: $h_{1,1}$, $h_{p1,1}$, $h_{2,1} = \gamma_{w2,1} \cap \gamma_{t2,0}$ and $h_{p2,1}$ symmetrical to $h_{2,1}$ towards $(y, z)$.

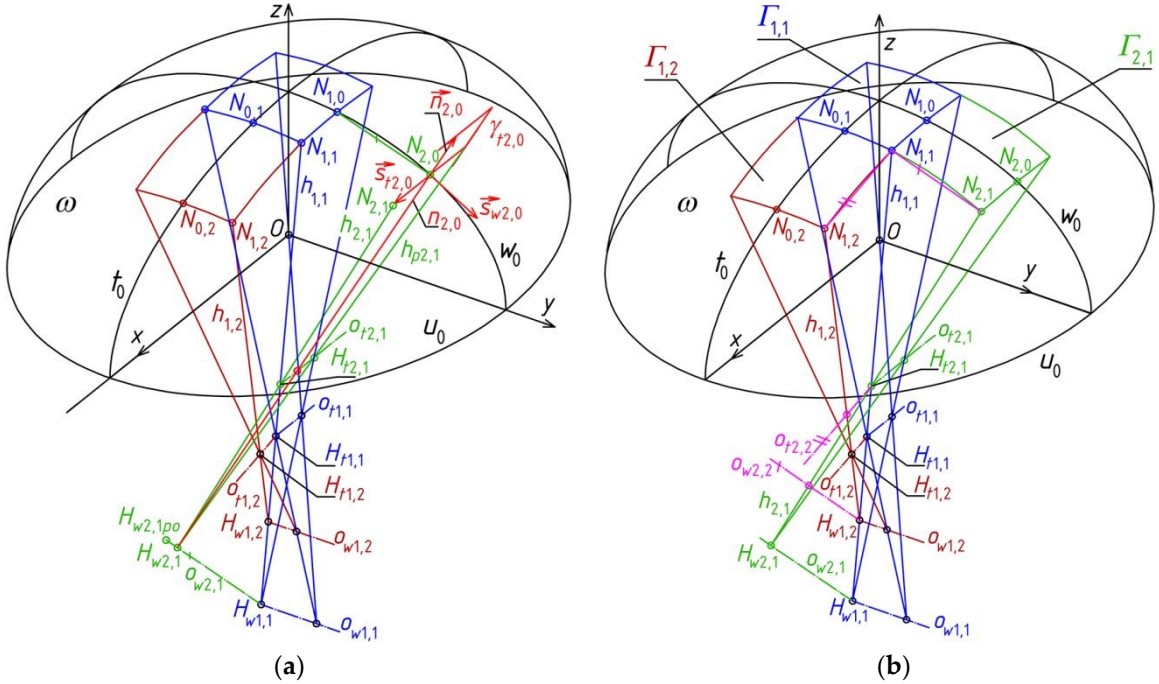

(a)           (b)

**Figure 15.** Constructions of a free form structure with the help of few reference tetrahedrons; (**a**) the auxiliary plane $\gamma_{t2,0}$ of $\Gamma_{2,1}$; (**b**) the reference tetrahedron $\Gamma_{2,1}$.

Reference tetrahedron $\Gamma_{2,2}$ is created in the way (Figure 16a) slightly different from the way used for $\Gamma_{1,2}$ and $\Gamma_{2,1}$ because two planes $\gamma_{w2,1}$ and $\gamma_{t1,2}$ from among the four planes of $\Gamma_{2,2}$ and three side edges $h_{1,1}$, $h_{1,2}$ and $h_{2,1}$ of $\Gamma_{2,2}$ have been obtained. In order to construct the fourth side edge $h_{2,2}$ of $\Gamma_{2,2}$ the following action should be executed. Straight line $o_{t2,2}$ is led through point $H_{t2,1} = h_{1,1} \cap h_{2,1}$ as parallel to straight line $(N_{1,1}, N_{1,2})$, where $N_{1,2} = h_{1,2} \cap \omega$. Straight line $o_{w2,2}$ is led through point $H_{w1,2} = h_{1,1} \cap h_{1,2}$ parallel to straight line $(N_{1,1}, N_{2,1})$, where $N_{2,1} = h_{2,1} \cap \omega$. Finally, points $H_{t2,2} = o_{t2,2} \cap h_{1,2}$ and $H_{w2,2} = o_{w2,2} \cap h_{2,1}$ determine edge $h_{2,2}$ (Figure 16b). Here, $h_{2,2}$ together with $h_{1,2}$ and $h_{2,1}$ determine two planes of $\Gamma_{2,2}$ being sought.

Four reference tetrahedrons $\Gamma_{i,j}$ (for $i$, $j$ = 1,2) were constructed so far. The other five reference tetrahedrons of $\Gamma$ (Figure 16b) can be obtained by transforming the above four tetrahedrons $\Gamma_{i,j}$ (for $i$, $j$ = 1,2) so that ($x$, $z$) and ($y$, $z$) ary the symmetry planes of $\Gamma$.

The final roof shell structure $\Omega$ composed of nine shell sectors $\Omega_{i,j}$ contained in nine $\Gamma_{i,j}$ is constructed. The activities leading to the determination of shell structure $\Omega$, being the sum of sectors $\Omega_{i,j}$ of ruled surfaces created on the basis of $\Gamma_{i,j}$ and $\Delta_{i,j}$ are similar to those ones presented earlier for single sector $\Omega_{1,1}$ contained in $\Gamma_{1,1}$. Ruled surfaces $\Omega_{i,j}$ are created on the basis of $\omega$ and positioned symmetrically towards ($x$, $z$) or ($y$, $z$) by analogy with the example described earlier. Roof structure $\Omega$ is shown in Figure 16b.

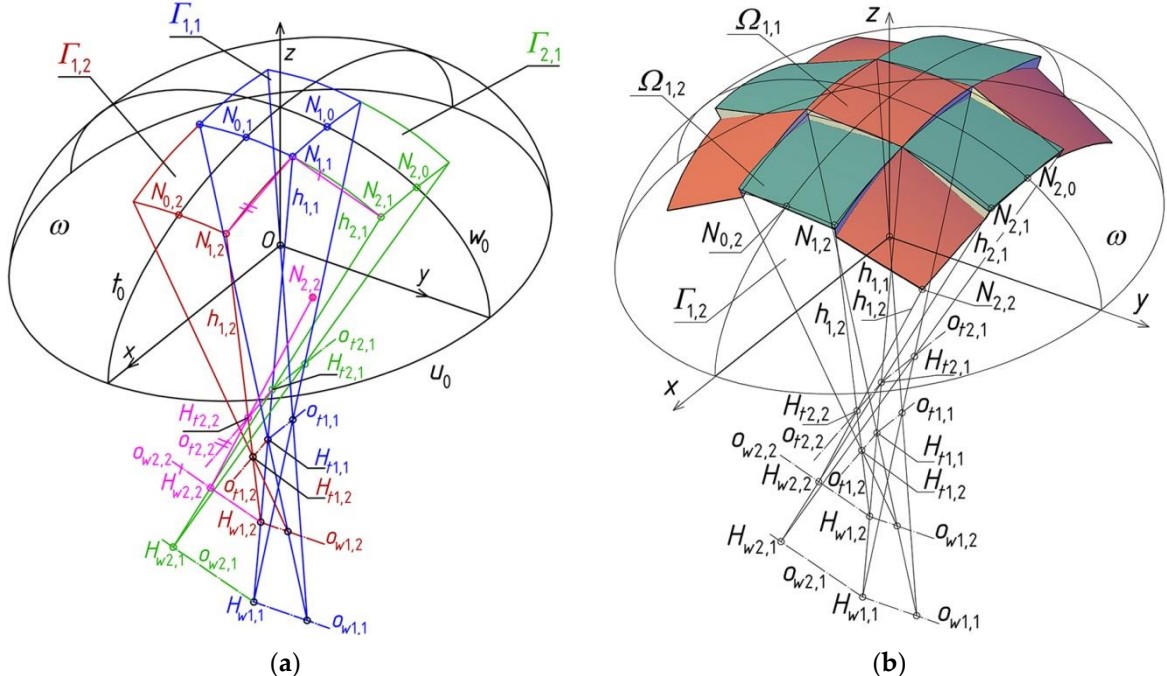

| (a) | (b) |

**Figure 16.** A free form structure created with the help of nine reference tetrahedrons: (**a**) Constructions of the reference tetrahedron $\Gamma_{2,2}$; (**b**) the simplified model of a shell roof structure.

Visualization of the achieved free form structure roofed with multi-segment shell structures is shown in Figure 17. It is possible to obtain many diversified, consistent architectural forms of such buildings from which two are presented in Figures 18 and 19. They are modifications of the structure created previously.

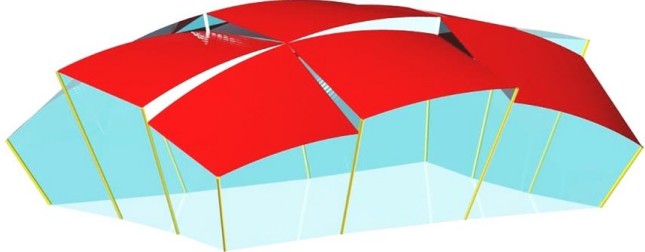

**Figure 17.** Visualization of the achieved free form structure roofed with a multi-segment shell structure.

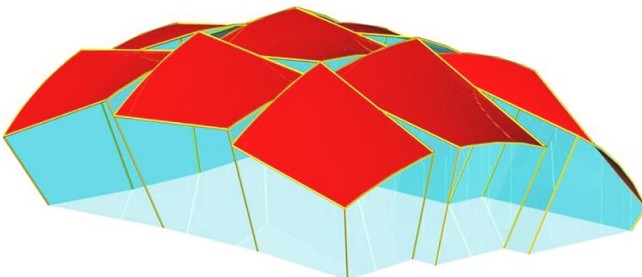

**Figure 18.** Visualization of the modified free form structure roofed with a multi-segment shell structure and folded elevation.

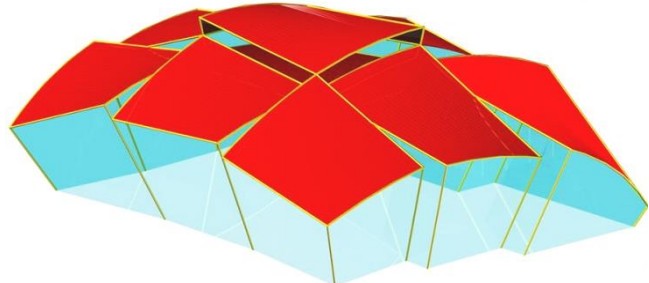

**Figure 19.** Visualization of another modified free form structure roofed with a multi-segment shell structure and folded elevation.

## 7. Optimized Structures Based on Regular Reference Surface

A non-rotational ellipsoid is used as a reference surface in the example presented below. The equation of this ellipsoid $\sigma$ is the same as previously (1), but $a = 24,000$ mm, $b = 18,000$ mm, $c = 11,000$ mm. The gable wall of the designed structure can be located in one plane (Figure 20a) or divided into two planar pieces symmetrical towards the plane $(x, z)$ (Figure 20b). The both forms are presented in the first part of the section. The shell roofs corresponding to these forms take very simple shapes. The roofs are created as the sums of a few shell strips whose directrices became the ellipses of the intersection of the gable wall plane or other almost vertical planes with the aforementioned reference ellipsoid $\sigma$.

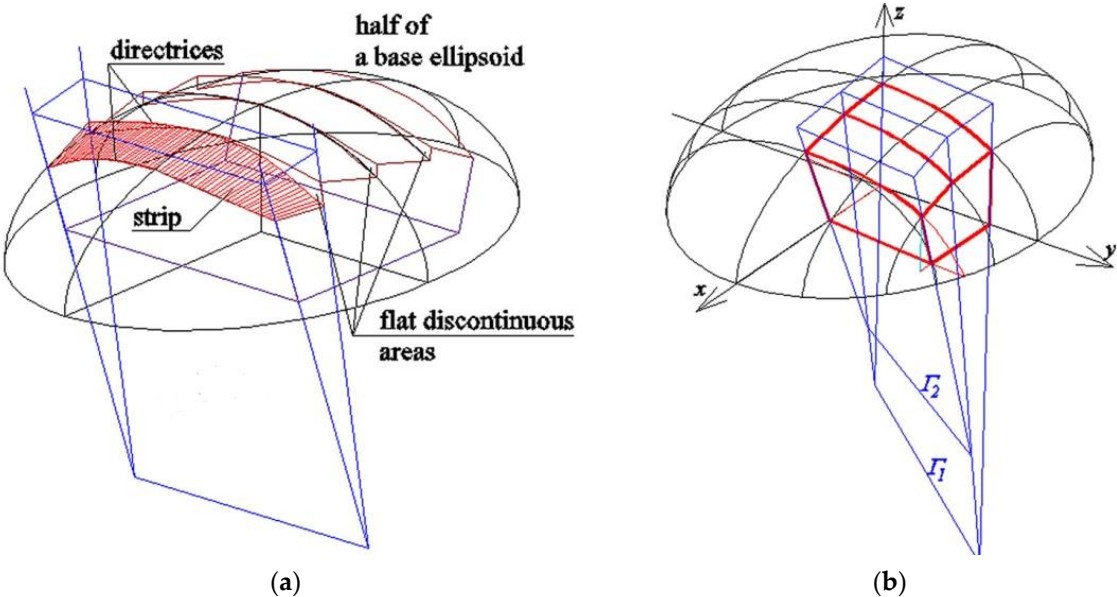

(**a**)    (**b**)

**Figure 20.** Geometric shaping of shell structures with a reference ellipsoid and reference tetrahedrons: (**a**) a single flat gable wall; (**b**) a symmetrical part of a gable wall.

The discussed method proposes to create more extended forms based on the reference surface $\sigma$, covered with compound shell structures supported by walls formed from many planar or shell segments, as shown in Figure 21. As a result, innovative, attractive and integrated building forms can be provided.

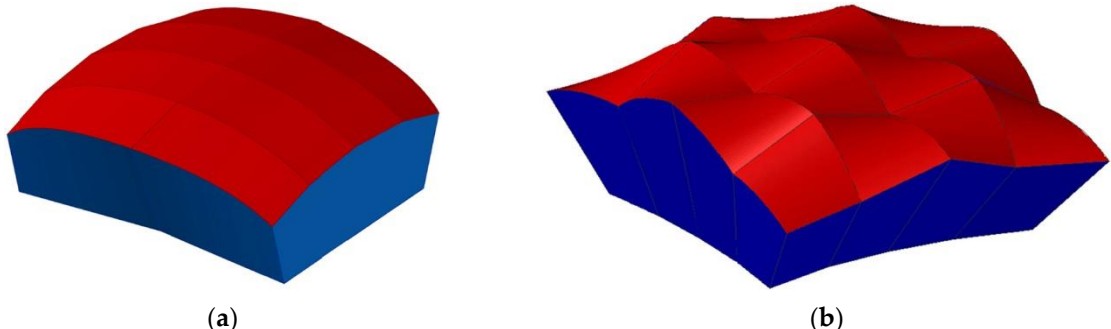

|         (a)          |          (b)          |

**Figure 21.** (**a**) Shell roof strips and the oblique gable wall composed of two parts contained in various planes; (**b**) Shell roof segments and the oblique gable wall composed of three parts contained in various planes.

If the number of the planar pieces of the gable wall is increased, then the integration of this wall with the entire building may be improved. In the next example, a regular reference structure is created. For that purpose, a finite number of points $N_{i,j}$ is defined on the reference ellipsoid with the help of ellipses $w_i$ and $t_i$ ($i = 0, 1, 2$) contained in vertical planes (Figure 22). The coordinates of the considered points $N_{i,j}$ ($i = 0, 1, 2, j = 0, 1, 2$) selected on the reference surface $\sigma$ are included in Table 6.

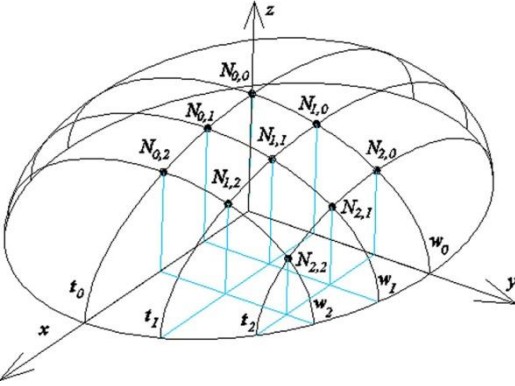

**Figure 22.** Visualization of the achieved free form structure roofed with a multi-segment shell structure.

**Table 6.** Coordinates of points $N_{i,j}$ selected on reference ellipsoid $\sigma$.

| Vertex | X-Coordinate | Y-Coordinate | Z-Coordinate |
|--------|--------------|--------------|--------------|
| $N_{0,0}$ | 0.0 | 0.0 | 11,000.0 |
| $N_{1,0}$ | 0.0 | 6459.0 | 10,267.0 |
| $N_{2,0}$ | 0.0 | 12,515.0 | 7906.0 |
| $N_{0,1}$ | 6487.0 | 0.0 | 10,590.0 |
| $N_{0,2}$ | 12,855.0 | 0.0 | 9289.0 |
| $N_{1,1}$ | 6487.0 | 6459.0 | 9828.0 |
| $N_{2,1}$ | 6487.0 | 12,515.0 | 7326.0 |
| $N_{1,2}$ | 12,855.0 | −6792.8 | 8409.0 |
| $N_{2,2}$ | 12,855.0 | 6459.0 | 5272.0 |

The parametric equations of the considered ellipses $t_i$ of reference ellipsoid $\sigma$ are given by:

$$
\begin{aligned}
x &= a_t \cdot \cos(\tau_i) \\
y &= y_{Ni,0} \\
z &= c_t \cdot \sin(\tau_i),
\end{aligned}
\tag{2}
$$

where $a_t = a \cdot \sqrt{1 - \frac{y_{Ni,0}^2}{b^2}}$, $c_t = c \cdot \sqrt{1 - \frac{y_{Ni,0}^2}{b^2}}$, $\tau_i$—the interdependent variable, $y_{Ni,0}$—$y$-coordinate of point $N_{i,0}$ ($i = 0, 1, 2$). The parametric equation of the ellipses $w_j$ of $\sigma$ are given by:

$$
\begin{aligned}
x &= x_{N0,j} \\
y &= b_w \cdot \cos(\omega_j) \\
z &= c_w \cdot \sin(\omega_j),
\end{aligned}
\tag{3}
$$

where $b_w = b \cdot \sqrt{1 - \frac{x_{N0,j}^2}{a^2}}$, $c_w = c \cdot \sqrt{1 - \frac{x_{N0,j}^2}{a^2}}$, $\omega_j$ - the interdependent variable, $x_{Ni,0} - x$-coordinate of point $N_{0,j}$ ($j = 0, 1, 2$).

One plane of the reference network can be created for each pair of two subsequent lines $\{n_{i,j}, n_{i+1,j}\}$ normal to $\sigma$ (Figure 23) to increase the integration degree of the entire structure with reference surface $\sigma$. On the basis of the above equations, straight lines $s_{tNi,j}$ and $s_{wNi,j}$ tangent to ellipses $t_i$ and $w_j$ of $\sigma$ at $N_{i,j}$ are determined, (Figure 23a). Based on these tangents, the directional vectors of straight lines $n_{i,j}$ normal to this ellipsoid at $N_{i,j}$ were calculated. The values of the components $[l_{ni,j}, m_{ni,j}, m_{ni,j}]$ of these directional vectors are given in Table 7.

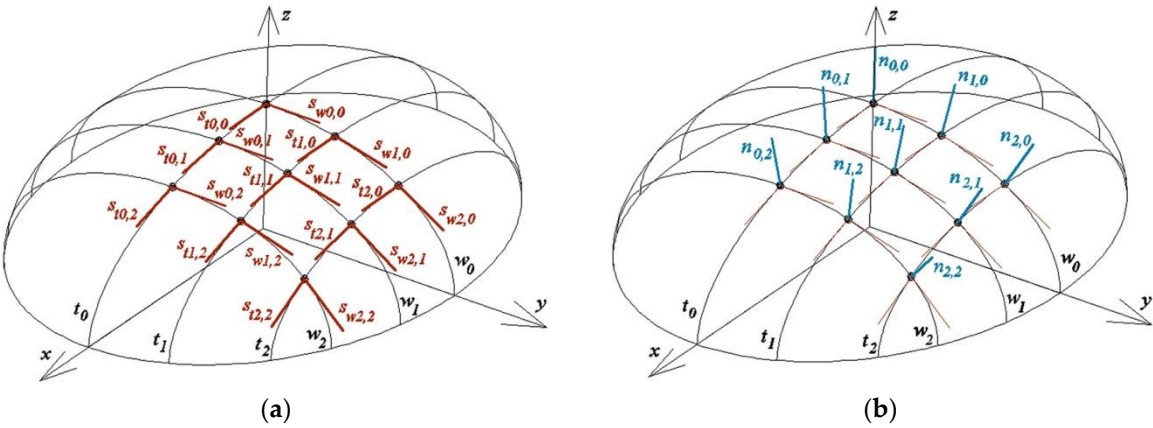

**Figure 23.** (**a**) Straight lines $s_{ti,j}$, $s_{wi,j}$ tangent to the reference ellipsoid at points spaced on its directrices $t_i$, $w_j$ at constant distances; (**b**) Straight lines $n_{i,j}$ normal to the reference ellipsoid at the same points.

**Table 7.** Values of the components of the directional vectors of straight lines $n_{i,j}$ normal to the ellipsoid.

| Vertex | $l_{ni,j}$ | $m_{ni,j}$ | $n_{ni,j}$ |
|---|---|---|---|
| $N_{0,0}$ | 0.0 | 0.0 | 5000.0 |
| $N_{1,0}$ | 0.0 | 1143.4 | 4867.5 |
| $N_{2,0}$ | 0.0 | 2544.3 | 4304.2 |
| $N_{0,1}$ | 638.1 | 0.0 | 4959.1 |
| $N_{0,2}$ | 1395.8 | 0.0 | 4801.2 |
| $N_{1,1}$ | 667.3 | 1181.0 | 4812.4 |
| $N_{2,1}$ | 774.6 | 2656.6 | 4164.4 |
| $N_{1,2}$ | 1474.8 | 1317.3 | 4592.3 |
| $N_{2,2}$ | 1789.5 | 3097.1 | 3493.6 |

In order to construct the first tetrahedron $\Gamma_{1,1}$ of reference network $\Gamma$, whose side edges $H_{i-1,1}$ ($i$ = 1–4) pass through points $N_{0,0}$, $N_{1,0}$, $N_{0,1}$, $N_{1,1}$, four pairs of straight lines $\{n_{0,0}, n_{0,1}\}$, $\{n_{0,1}, n_{1,1}\}$, $\{n_{1,0}, n_{1,1}\}$ and $\{n_{0,0}, n_{1,0}\}$ normal to $\sigma$ at these points must be considered. Next, for each pair of these normals, the intersection of both lines with a straight line perpendicular to them should be constructed. For the pair $\{n_{0,1}, n_{1,1}\}$, there is a straight line $n_{Hw1,1}$ and points $H_{w0,1\_1,1} \in n_{0,1}$ and $H_{w,11\_1,1} \in n_{1,1}$. For the pair $\{n_{1,0}, n_{1,1}\}$ there is the straight line $n_{Ht1,1}$ and points $H_{t1,0\_1,1} \in n_{1,0}$ and $H_{t1,1\_1,1} \in n_{1,1}$ (Figure 24).

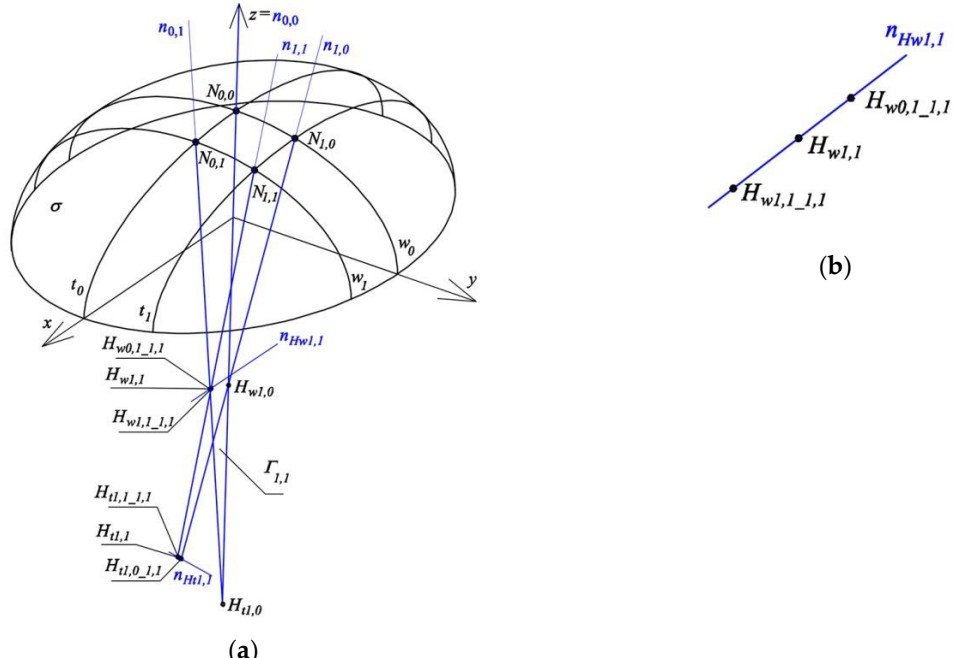

**Figure 24.** (**a**) Creation of the first reference tetrahedron $\Gamma_{1,1}$: (**a**) side edges and vertices, (**b**) search for one vertex $H_{w1,1}$.

For the other two cases: $\{n_{0,0}, n_{0,1}\}$, $\{n_{0,0}, n_{1,0}\}$, both straight lines of each of these pairs are coplanar, and therefore these lines cross each other at vertices $H_{w0,0}$ and $H_{t0,0}$ of the reference tetrahedron $\Gamma_{1,1}$, and it is not necessary to carry out relevant constructions. The notation of point $H_{w0,1\_1,1}$ should be interpreted as follows: (a) symbol $w$ indicates that this element is related to curve $w_j$; (b) subscript 0,1 indicates that this point is referred to $N_{0,1}$; and (c) subscript 1,1 means that the considered point is used for determining $\Gamma_{1,1}$.

A tetrad of planes $\zeta_{tN0,1}$, $\zeta_{wN1,0}$, $\zeta_{tN1,1}$, $\zeta_{wN1,1}$ of tetrahedron $\Gamma_{1,1}$ is formed from two planes ($x$, $z$), ($y$, $z$) of the orthogonal coordinate system [$x$,$y$,$z$] and two planes defined by the following triads of points: ($N_{1,0}$, $N_{1,1}$, $H_{t1,1}$), ($N_{0,1}$, $N_{1,1}$, $H_{w1,1}$), where $H_{t1,1}$ is the middle point of section $H_{t1,0\_1,1}H_{t1,1\_1,1}$, however, $H_{w1,1}$ is the middle point of section $H_{w0,1\_1,1}H_{w1,1\_1,1}$. Ultimately, edges $h_{0,0\_1,1} = \zeta_{tN0,1} \cap \zeta_{wN1,0}$, $h_{0,1\_1,1} = \zeta_{tN0,1} \cap \zeta_{wN1,1}$, $h_{1,0\_1,1} = \zeta_{tN1,1} \cap \zeta_{wN1,0}$, $h_{1,1\_1,1} = \zeta_{tN1,1} \cap \zeta_{wN1,1}$ are constructed. They are preliminary approximations of the side edges of the polyhedral reference structure $\Gamma$, and take positions close to the positions of straight lines $n_{i,j}$ normal to the arbitrary reference ellipsoid $\sigma$ at points $N_{i,j}$ ($i$ = 0,1, $j$ = 0,1).

Each two subsequent straight lines from $\{h_{0,0\_1,1}, h_{0,1\_1,1}, h_{1,0\_1,1}, h_{1,1\_1,1}\}$ passing through adjacent points $N_{i,j}$ of ellipsoid $\sigma$ intersect at four corresponding vertices $\{H_{w1,0}, H_{t1,0}, H_{w1,1}, H_{t1,1}\}$ of the tetrahedron $\Gamma_{1,1}$. Points $N_{i,j}$ ($i$ = 0,1, $j$ = 0,1) together with vertices $N_{i,j}$ define the four planes $\zeta_{tN0,1}$, $\zeta_{wN1,0}$, $\zeta_{wN1,1}$ and $\zeta_{tN1,1}$ of $\Gamma_{1,1}$.

In order to construct the second tetrahedron $\Gamma_{2,1}$, whose side edges pass through points $N_{1,0}$, $N_{1,1}$, $N_{2,1}$, $N_{2,0}$ (Figure 25), it is necessary to take into account the previously considered pair $\{n_{1,0}, n_{1,1}\}$ and next three pairs of straight lines $\{n_{2,0}, n_{2,1}\}$, $\{n_{1,1}, n_{2,1}\}$ and $\{n_{1,0}, n_{2,0}\}$.

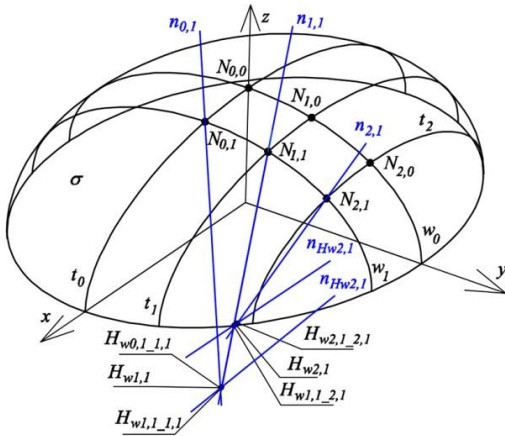

**Figure 25.** Creation of the second reference tetrahedron $\Gamma_{2,1}$.

Next, for each of these pairs, the points of both straight lines intersecting with a straight line perpendicular to them should be constructed. In the case of pair $\{n_{1,1}, n_{2,1}\}$, these are points $H_{w1,1\_2,1}$ $\in n_{1,1}$ and $H_{w,21\_2,1} \in n_{2,1}$ and straight line $n_{Hw2,1}$ (Figure 25). In the case of pair $\{n_{2,0}, n_{2,1}\}$, these are points $H_{t2,0\_2,1} \in n_{2,0}$ and $H_{t2,1\_2,1} \in n_{2,1}$. Straight lines $\{n_{1,0}, n_{2,0}\}$ are coplanar and intersect at vertex $H_{w2,0}$ of tetrahedron $\Gamma_{2,1}$. The side facets of tetrahedron $\Gamma_{2,1}$ are: $\zeta_{tN1,1}, \zeta_{wN2,0}, \zeta_{tN2,1}, \zeta_{wN2,1}$, where $\zeta_{wN2,0}$ is the plane $(y, z)$ of the coordinate system $[x, y, z]$, $\zeta_{tN1,1}$ was described earlier.

In turn, two planes: $\zeta_{tN2,1}, \zeta_{wN2,1}$ are defined respectively by the following triads of points: $(N_{2,0}, N_{2,1}, H_{t2,1})$, $(N_{1,1}, N_{2,1}, H_{w2,1})$, where $H_{t2,1}$ is the middle of section $H_{t2,0\_2,1}H_{t2,1\_1,1}$, and $H_{w2,1}$ is the middle of segment $H_{w1,1\_2,1}H_{w2,1\_2,1}$. Straight lines $h_{1,0\_2,1} = \zeta_{tN1,1} \cap \zeta_{wN2,0}$, $h_{1,1\_2,1} = \zeta_{tN1,1} \cap \zeta_{wN2,1}$, $h_{2,0\_2,1} = \zeta_{tN2,1} \cap \zeta_{wN2,0}$, $h_{2,1\_1,1} = \zeta_{tN2,1} \cap \zeta_{wN2,1}$ are initial approximations of straight lines $n_{Ni,j}$ normal to the reference surface $\sigma$ at points $N_{i,j}$ ($i = 1, 2, j = 0, 1$) and are taken as preliminary approximations of the four side edges of tetrahedron $\Gamma_{2,1}$. Each two subsequent straight lines from the four following lines $\{h_{1,0\_2,1}, h_{1,1\_2,1}, h_{2,0\_2,1}, h_{2,1\_1,1}\}$ pass through adjacent points $N_{i,j}$ of ellipsoid $\sigma$ and intersect in the appropriate four vertices $\{H_{w2,0}, H_{t2,0}, H_{w2,1}, H_{t2,1}\}$ of tetrahedron $\Gamma_{2,1}$ (Figure 26).

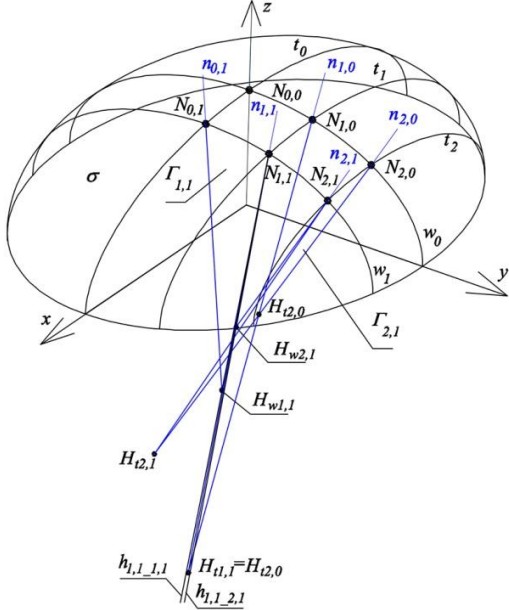

**Figure 26.** Creation of the sum of two adjacent tetrahedrons $\Gamma_{1,1}$, $\Gamma_{2,1}$ does not give a reference structure.

The sum of constructed tetrahedrons $\Gamma_{1,1}$ and $\Gamma_{2,1}$ does not create reference structure $\Gamma$, because, for example, straight lines $h_{1,1\_1,1}, \subset \Gamma_{1,1}$, i $h_{1,1\_2,1} \subset \Gamma_{2,1}$ are not identical (Figure 26), so the above

tetrahedrons do not have a common side edge passing through point $N_{1,1}$. In this case, the set of three adjacent planes $\zeta_{tN1,1} \circ \zeta_{wN1,1} \circ \zeta_{wN2,1}$ does not have a common edge, so the above tetrahedrons do not share any common wall.

Therefore, some operation is necessary to replace the two above various straight lines with one straight line in an effective way, so that the above system was replaced by a new system of three planes $\zeta_{tN1,1n} \circ \zeta_{wN1,1n} \circ \zeta_{wN2,1n}$ having one common edge $k_{1,1}$. However, the position of this edge of the new system should be the closest possible to the position of normal $n_{1,1}$ of ellipsoid $\sigma$ at point $N_{1,1}$. Therefore, an optimization process is necessary, so that sum $S_{Min}$ of square of angles $\varphi_{i,j}$ between each plane $\zeta_{tNi,jn}$ or $\zeta_{wi,jn}$ of the new system and the corresponding plane $\zeta_{tNi,jn}$ or $\zeta_{wi,jn}$ of the old system was the smallest possible. The optimization condition reads:

$$\sum_{i=1}^{i_n=2} \sum_{j=1}^{j_n=2} \varphi_{Hri,jn}^2 \, S_{Min} \; = \; min, \tag{4}$$

where: $r = w$ or $t$, so it is obtained $\varphi_{i,jn}$ for points $H_{wi,jn}$ and $\zeta_{wi,jn}$, and $\varphi_{i,jn}$ for points $H_{ti,jn}$ and $\zeta_{tNi,jn}$.

It is then assumed that edge $k_{1,1}$ of the three new planes will be the closest possible to the line $n_{1,1}$ normal to ellipsoid $\sigma$ within acceptable modeling accuracy. The description of the way of determining straight line $k_{1,1}$ will be presented after considering two next reference tetrahedrons $\Gamma_{1,2}$, $\Gamma_{2,2}$, because this straight line will replace four corresponding side edges of tetrahedrons $\Gamma_{1,1}$, $\Gamma_{2,1}$, $\Gamma_{1,2}$, $\Gamma_{2,2}$ with no common side edge passing through point $N_{1,1}$. The results of the optimization process performed for the reference structure $\Gamma_n$ composed of four reference tetrahedrons $\Gamma_{i,jn}$ replacing $\Gamma_{i,j}$ ($i = 1.2$, $j = 1.2$) are also presented at the end of this section.

The last two considered tetrahedrons $\Gamma_{2,1}$, $\Gamma_{2,2}$ are defined in a manner analogous to the one presented earlier for $\Gamma_{1,1}$. Therefore, the following pairs of straight lines are investigated: $\{n_{0,2}, n_{1,2}\}$, $\{n_{1,1}, n_{1,2}\}$, $\{n_{1,2}, n_{2,2}\}$, $\{n_{2,1}, n_{2,2}\}$ and their intersecting points with a corresponding straight line perpendicular to each pair are sought. In the case of pair $\{n_{0,2}, n_{1,2}\}$, these are points $H_{w0,2\_1,2} \in n_{0,2}$ and $H_{w1,2\_1,2} \in n_{1,2}$. In the case of pair $\{n_{1,1}, n_{1,2}\}$, these are points $H_{t1,1\_1,2} \in n_{1,1}$ i $H_{t1,2\_1,2} \in n_{1,2}$ (Figure 27a). In the case of pair $\{n_{1,2}, n_{2,2}\}$, these are points $H_{w1,2\_2,2} \in n_{1,2}$ i $H_{w2,2\_1,2} \in n_{2,2}$. For pair $\{n_{2,1}, n_{2,2}\}$, these are points $H_{t2,1\_2,2} \in n_{2,1}$ i $H_{t2,2\_2,2} \in n_{2,2}$.

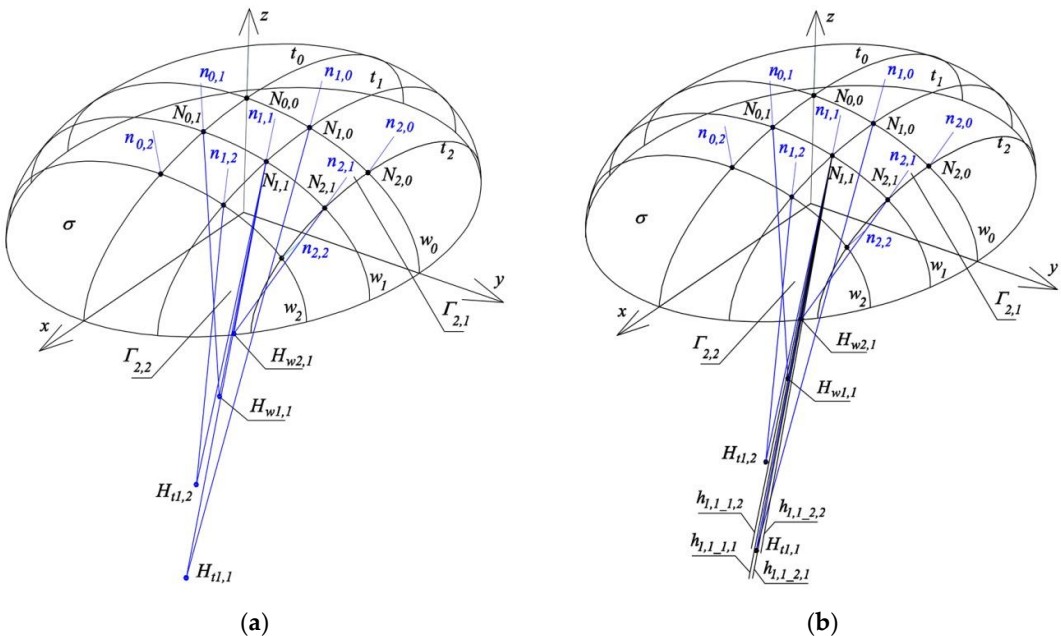

**Figure 27.** Construction of side edges of four reference tetrahedrons $\Gamma_{1,1}$, $\Gamma_{2,1}$, $\Gamma_{1,2}$, $\Gamma_{2,2}$: (**a**) various side edges; (**b**) search for one common side edge.

Tetrahedron $\Gamma_{1,2}$ is created by means of planes $\zeta_{tN0,2}$, $\zeta_{tN1,2}$, $\zeta_{wN1,2}$, $\zeta_{wN1,1}$, where $\zeta_{tN0,2}$ is the plane $(x, z)$ of the coordinate system, $\zeta_{wN1,1}$ was defined before, while two planes $\zeta_{tN1,2}$, $\zeta_{wN1,2}$ are defined by the following triads of points: $(N_{1,1}, N_{1,2}, H_{t1,2})$, $(N_{0,2}, N_{1,2}, H_{w1,2})$, where $H_{t1,2}$ is the middle of $H_{t1,1\_1,2}H_{t1,2\_1,2}$ section, $H_{w1,2}$ is the middle of $H_{w0,2-\_1,2}H_{w1,2-\_1,2}$. Straight lines $h_{0,1\_1,2} = \zeta_{tN0,2} \cap \zeta_{wN1,1}$, $h_{0,2\_1,2} = \zeta_{tN0,2} \cap \zeta_{wN1,2}$, $h_{1,1\_1,2} = \zeta_{tN1,2} \cap \zeta_{wN1,1}$, $h_{1,2\_1,2} = \zeta_{tN1,2} \cap \zeta_{wN1,2}$ are adopted as the edges of tetrahedron $\Gamma_{1,2}$. These edges take positions close to the positions of straight lines $n_{i,j}$ normal to reference surface $\sigma$ at points $N_{i,j}$ (for $i = 0,1$, $j = 1,2$). However, they are not side edges of the searched polyhedral reference structure because $\Gamma_{1,2}$ has no common edge with $\Gamma_{1,1}$ or $\Gamma_{2,1}$.

Tetrahedron $\Gamma_{2,2}$ is defined by means of planes $\zeta_{tN1,2}$, $\zeta_{tN2,2}$, $\zeta_{wN2,1}$, $\zeta_{wN2,2}$, where $\zeta_{tN1,2}$ and $\zeta_{wN2,1}$ were defined before. The next two planes $\zeta_{tN2,2}$, $\zeta_{wN2,2}$ should be determined using the following triplets of points: $(N_{2,1}, N_{2,2}, H_{t2,2})$, $(N_{1,2}, N_{2,2}, H_{w2,2})$, where $H_{t2,2}$ is the middle of segment $H_{t2,1\_2,2}H_{t2,2\_2,2}$, while $H_{w2,}$ is the middle of segment $H_{w1,2-\_2,2}H_{w2,2-\_2,2}$. The following straight lines $h_{1,1\_2,2} = \zeta_{tN1,2} \cap \zeta_{wN2,1}$, $h_{1,2\_2,2} = \zeta_{tN1,2} \cap \zeta_{wN2,2}$, $h_{2,1\_2,2} = \zeta_{tN2,2} \cap \zeta_{wN2,1}$, $h_{2,2\_2,2} = \zeta_{tN2,2} \cap \zeta_{wN2,2}$ are adopted as the edges of reference of tetrahedron $\Gamma_{2,2}$ and are the initial approximation of the edges of the polyhedral reference structure $\Gamma$.

The structure created as a result of adding the reference tetrahedrons $\Gamma_{1,1}$, $\Gamma_{2,1}$, $\Gamma_{1,2}$, $\Gamma_{2,2}$ is not reference structure $\Gamma$ because, similarly as for the case of the sum of tetrahedrons $\Gamma_{1,1}$, $\Gamma_{2,1}$ considered earlier, two triads of planes: $\zeta_{tN1,2} \circ \zeta_{wN1,2} \circ \zeta_{wN2,2}$, $\zeta_{tN2,1} \circ \zeta_{wN2,1} \circ \zeta_{tN2,2}$ and four planes $\zeta_{tN1,1} \circ \zeta_{wN1,1} \circ \zeta_{wN2,1} \circ \zeta_{tN1,2}$ do not have common side edges. Thus, tetrahedrons $\Gamma_{1,2}$, $\Gamma_{2,2}$ and tetrahedrons $\Gamma_{2,1}$, $\Gamma_{2,2}$ as well as $\Gamma_{1,1}$, $\Gamma_{2,1}$, $\Gamma_{1,2}$, $\Gamma_{2,2}$ do not have appropriate common side edges passing through point $N_{2,1}$ or $N_{1,2}$ or $N_{1,1}$ (Figure 27b). Therefore, it is necessary to replace the above two triads and one tetrad with two new triads $\zeta_{tN1,2n} \circ \zeta_{wN1,2n} \circ \zeta_{wN2,2n}$, $\zeta_{tN2,1n} \circ \zeta_{wN2,1n} \circ \zeta_{tN2,2n}$ and one new tetrad $\zeta_{tN1,1n} \circ \zeta_{wN1,1n} \circ \zeta_{wN2,1n} \circ \zeta_{tN1,2n}$ in such a way that the position of the new planes and their edges is close to the positions of the respective straight lines $n_{i,j}$ normal to reference surface $\sigma$ with the most possible precision.

Therefore, a second step of the initiated process of replacing straight lines $h_{r,s\_i,j}$ $(i, j, r, t = 1, 2)$ of planes $\zeta_{tNi,j}$ or $\zeta_{wNi,j}$ belonging to the created tetrahedrons $\Gamma_{i,j}$ not producing the polyhedral reference structure $\Gamma$, with side edges $k_{i,j}$ of planes $\zeta_{tNi,jn}$ and $\zeta_{wNi,jn}$ of the searched polyhedral reference structure $\Gamma_n$ of several $\Gamma_{i,jn}$, is necessary. Since the above straight lines and planes should be as close as possible to straight lines $n_{i,j}$ normal to reference surface $\sigma$, it was assumed that sum $S_{Min}$ (4) of squares of angles between the planes of the achieved new systems of planes $\zeta_{tN1,2n} \circ \zeta_{wN1,2n} \circ \zeta_{wN2,2n}$, $\zeta_{tN2,1n} \circ \zeta_{wN2,1n} \circ \zeta_{tN2,2n}$, $\zeta_{tN1,1n} \circ \zeta_{wN1,1n} \circ \zeta_{wN2,1n} \circ \zeta_{tN1,2n}$ and the corresponding planes of the previously created old systems of planes $\zeta_{tN1,2} \circ \zeta_{wN1,2} \circ \zeta_{wN2,2}$, $\zeta_{tN2,1} \circ \zeta_{wN2,1} \circ \zeta_{tN2,2}$, $\zeta_{tN1,1} \circ \zeta_{wN1,1} \circ \zeta_{wN2,1} \circ \zeta_{tN1,2}$ should be the smallest possible.

To calculate the angles between the $\zeta_{tNi,j}$, $\zeta_{wNi,j}$ planes determined in the first step for the tetrahedrons $\Gamma_{i,j}$ and $\zeta_{tNi,jn}$, $\zeta_{wNi,jn}$ for the $\Gamma_{i,jn}$ reference tetrahedrons forming the meshes of the searched reference network $\Gamma_n$, and estimated in the second step of the algorithm of the presented method the following formula was used

$$\varphi_{Hri,jn} = \frac{\pi}{2} - \text{asin}\left(\frac{n_{ki,j} \cdot n_{ki,jn}}{\left|n_{ki,j}\right| \cdot \left|n_{ki,jn}\right|}\right). \tag{5}$$

where $n_{ki,j}$ is the unit vector normal to plane $\zeta_{tNi,j}$ or $\zeta_{wNi,j}$, $n_{ki,jn}$ is the unit directional vector of $\zeta_{tNi,jn}$ or $\zeta_{wNi,jn}$, $\Pi = 3.14159$.

As a result of the optimization process, the following objects were obtained (Figure 28): (a) edge $k_{1,2}$ of three new planes $\zeta_{tN1,2n} \circ \zeta_{wN1,2n} \circ \zeta_{wN2,2n}$, which replaces two straight lines $h_{1,2\_1,2} \subset \Gamma_{1,2}$, $h_{1,2\_2,2} \subset \Gamma_{1,2}$ as accurately as possible; (b) edge $k_{2,1}$ of three new planes $\zeta_{tN2,1n} \circ \zeta_{wN2,1n} \circ \zeta_{tN2,2n}$, which replaces two straight lines $h_{2,1\_2,1} \subset \Gamma_{2,1}$, $h_{2,1\_2,2} \subset \Gamma_{2,2}$ as precisely as possible; and (c) edge $k_{1,1}$ of four new planes $\zeta_{tN1,1n} \circ \zeta_{wN1,1n} \circ \zeta_{wN2,1n} \circ \zeta_{tN1,2n}$, which substitutes four straight lines $h_{1,1\_1,1} \subset \Gamma_{1,1}$, $h_{1,1\_1,2} \subset \Gamma_{1,2}$, $h_{1,1\_2,1} \subset \Gamma_{2,1}$, $h_{1,1\_2,2} \subset \Gamma_{2,2}$ as exactly as possible. On the basis of the above

planes, it is possible to determine edge $k_{2,2}$ of $\Gamma_n$ as the straight line being the intersection of planes $\zeta_{wN2,2n}$, $\cap\ \zeta_{tN2,2n}$, where $\zeta_{tN2,2n}$ is defined by edge $k_{2,1}$ and point $N_{2,2}$, and $\zeta_{wN2,2n}$ is defined by edge $k_{1,2}$ and point $N_{2,2}$. As a result, the straight lines $k_{1,1}$, $k_{1,2}$, $k_{2,1}$, $k_{2,2}$, whose positions are very close to the positions of $n_{1,1}$, $n_{1,2}$, $n_{2,1}$, $n_{2,2}$ normal to ellipsoid $\sigma$, were obtained.

During the process of optimizing the positions of the facets and edges of reference tetrahedrons $\Gamma_{i,jn}$ to the positions of straight lines $n_{i,j}$ normal to the reference surface $\sigma$, the positions of the following points were changed: $H_{w1,1}$, $H_{w2,1}$, $H_{w2,2}$, $H_{t2,1}$ respectively on lines ($H_{w0,1\_1,1}$, $H_{w1,1\_1,1}$), ($H_{w1,1\_2,1}$, $H_{w2,1\_2,1}$), ($H_{w1,2-\_2,2}$, $H_{w2,2\_2,2}$), ($H_{t2,0\_2,1}$, $H_{t2,1\_2,1}$). As a result, their new positions $H_{w1,1n}$, $H_{w2,1n}$, $H_{w2,2n}$, $H_{t2,1n}$ made it possible to build triples of points ($N_{0,1}$, $N_{1,1}$, $H_{w1,1n}$), ($N_{1,1}$, $N_{2,1}$, $H_{w2,1n}$), ($N_{1,2}$, $N_{2,2}$, $H_{w2,2n}$), ($N_{2,0}$, $N_{2,1}$, $H_{t2,1n}$) determining the planes $\zeta_{wN1,1n}$, $\zeta_{wN2,1n}$, $\zeta_{wN2,2n}$, $\zeta_{tN2,1n}$ sought. The location of points $H_{w1,1}$, $H_{w2,1}$, $H_{w2,2}$, $H_{t2,1}$ was controlled by parameters $w_{Hw1,1n}$, $w_{Hw2,1n}$, $w_{Hw2,2n}$, $w_{Ht2,1n}$, that is the division coefficients of sections $H_{w0,1\_1,1}H_{w1,1\_1,1}$, $H_{w1,1\_2,1}H_{w2,1\_2,1}$, $H_{w1,2-\_2,2}H_{w2,2\_2,2}$ i $H_{t2,0\_2,1}H_{t2,1\_2,1}$, in contrast to the vertices $H_{w1,1}$, $H_{w2,1}$, $H_{w2,2}$, $H_{t2,1}$ adopted in the middles of the respective sections considered in the previous step. For other points $H_{wi,jn}$, $H_{ti,jn}$, the values of the division coefficients are the result of the optimization process and depend on the aforementioned four coefficients.

The algorithm of defining the optimal reference structure $\Gamma_n$ based on the obtained points $H_{w1,1}$, $H_{w2,1}$, $H_{w2,2}$, $H_{t2,1}$ is presented below (Figure 28). Structure $\Gamma_n$ is the sum of reference tetrahedrons $\Gamma_{i,jn}$, whose edges $k_{i,j}$ and planes $\zeta_{tNi,jn}$ can be achieved on the basis of the above four optimized division coefficients in the following way.

Points $N_{1,1}$ and $H_{w1,1n}$ define edge $k_{1,1}$. Straight line $k_{1,1}$ and point $N_{0,1}$ determine plane $\zeta_{wN1,1n}$ of the reference structure $\Gamma_n$. Analogously, the following planes: $\zeta_{wN2,1n}$, $\zeta_{tN1,1n}$, $\zeta_{tN1,2n}$ of structure $\Gamma_n$ pass through edge $k_{1,1}$ and points $N_{1,0}$, $N_{1,2}$, $N_{2,1}$, respectively. Similarly, plane $\zeta_{wN2,2n}$ passing through points $N_{1,2}$, $N_{2,2}$, $H_{w2,2n}$ is the plane of structure $\Gamma_n$. Edge $k_{1,2}$ of structure $\Gamma_n$ is the intersection of planes $\zeta_{tN1,2n}$, $\zeta_{wN2,2n}$. Edge $k_{1,2}$ and point $N_{0,2}$ define plane $\zeta_{wN1,2n}$. Planes $\zeta_{tN2,1n}$ and $\zeta_{wN2,1n}$ intersect each other in edge $k_{2,1}$. Plane $\zeta_{tN2,2n}$ is determined by points $N_{2,0}$, $N_{2,1}$ and $H_{t2,1n}$. Plane $\zeta_{wN2,2n}$ is determined by points $N_{1,2}$, $N_{2,2}$, $H_{w2,2n}$. Edge $k_{2,2}$ is the intersection of planes $\zeta_{tN2,2n}$ and $\zeta_{wN2,2n}$.

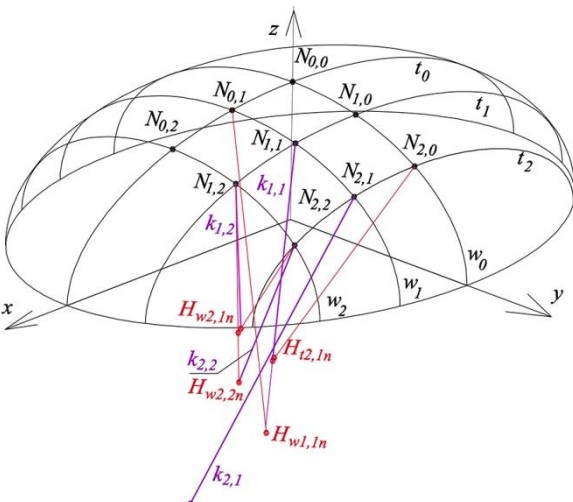

**Figure 28.** Creation of the side edges of reference structure $\Gamma_n$ composed of four reference tetrahedrons $\Gamma_{1,1n}$, $\Gamma_{2,1n}$, $\Gamma_{1,2n}$, $\Gamma_{2,2n}$.

Edges $k_{0,1}$ and $k_{0,2}$ of structure $\Gamma_n$ are the straight lines of the intersection of planes $\zeta_{wN1,1n}$, $\zeta_{wN1,2n}$ with plane ($x$, $z$). Edges $k_{1,0}$ and $k_{2,0}$ of $\Gamma_n$ are the straight lines of the intersection of planes $\zeta_{tN1,1n}$, $\zeta_{tN2,1n}$ with plane ($y$, $z$). Axis $z$ is adopted as edge $k_{0,0}$ of $\Gamma_n$. The reference tetrahedron $\Gamma_{2,2}$ (Figure 29) is created as the last part of the one-fourth of $\Gamma_n$.

The reference structure $\Gamma_n$ is the sum of all reference tetrahedrons $\Gamma_{i,jn}$ whose walls, contained in the aforementioned planes $\zeta_{wNi,jn}$, $\zeta_{tNi,jn}$ ($i$ = 0, 1, 2 and $j$ = 0, 1, 2), are common to each pair of the adjacent reference tetrahedrons, and side edges $k_{i,j}$ are the shared corners of pairs, triples or tetrads of the neighboring reference tetrahedrons. The results of the optimization process performed for one of the four quarters of the considered reference structure $\Gamma_n$ symmetrical towards two planes of coordinate system [$x$, $y$, $z$] are presented in Figure 30.

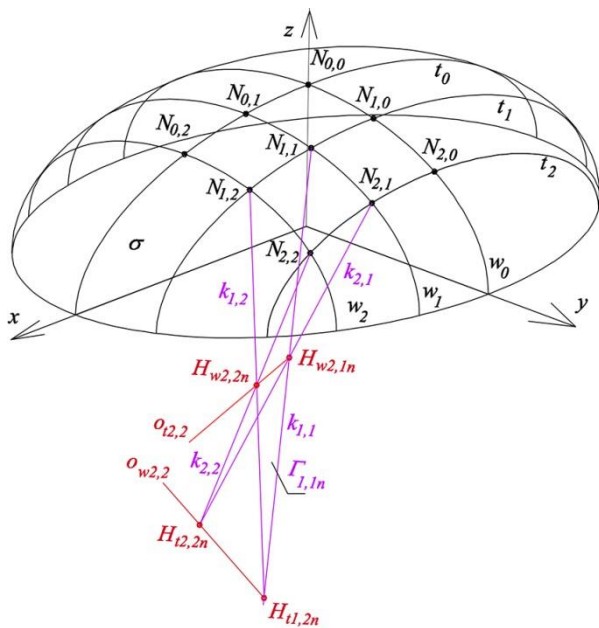

**Figure 29.** Creation of reference tetrahedron $\Gamma_{2,2n}$ of $\Gamma_n$ structure.

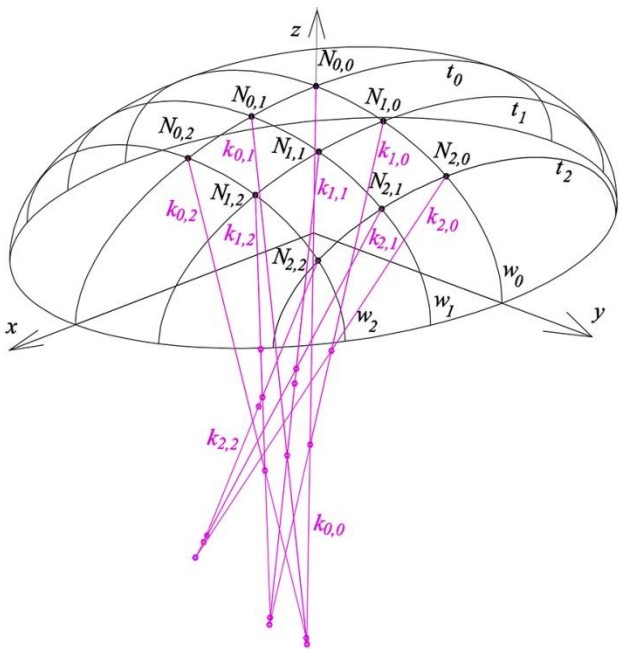

**Figure 30.** One-fourth of reference structure $\Gamma_n$, composed of four reference tetrahedrons $\Gamma_{1,1n}$, $\Gamma_{2,1n}$, $\Gamma_{1,2n}$, $\Gamma_{2,2n}$.

The values of selected coefficients $w_{Hwi,jn}$ or $w_{Hti,jn}$ evaluated in the presented iterative optimization process are included in Table 8. The obtained values of some angles between planes $\zeta_{tNi,j}$,

$\zeta_{wNi,j}$ of old tetrahedrons $\Gamma_{i,j}$ and planes $\zeta_{tNi,jn}$, $\zeta_{wNi,j}$ of new reference tetrahedrons $\Gamma_{i,jn}$ are included in Table 9. The investigated one-fourth of the structure $\Gamma_n$ is composed of four reference tetrahedrons $\Gamma_{i,jn}$ ($i$ = 1, 2 and $j$ = 1, 2). The values of components $l_{ki,j}$, $m_{ki,j}$, $m_{ki,j}$ of the directional vectors of side edges $k_{i,j}$ of structure $\Gamma_n$, passing through points $N_{i,j}$ are given in Table 10.

**Table 8.** Division coefficients subsequently accepted in the iterative optimization process.

| Iteration Step [No] | $w_{Hw1,1n}$ | $w_{Hw2,1n}$ | $w_{Hw2,2n}$ | $w_{Ht2,1n}$ |
|---|---|---|---|---|
| 1 | 0.50 | 0.50 | 0.50 | 0.50 |
| 2 | 1.00 | 0.60 | 0.50 | 0.50 |
| 3 | 2.00 | 0.70 | 0.50 | 0.50 |
| 4 | 3.00 | 0.80 | 0.50 | 0.50 |
| 5 | 4.55 | 0.95 | 0.50 | 0.50 |
| 6 | 4.55 | 0.95 | 0.20 | 0.50 |
| 7 | 4.58 | 0.95 | 0.21 | 0.50 |

**Table 9.** Decreasing values of sum $S_{Min}$ of square of angles $\varphi_{i,j}$ between planes $\zeta_{tNi,j}$, $\zeta_{wNi,j}$ of the old tetrahedrons $\Gamma_{i,j}$ and planes $\zeta_{tNi,jn}$, $\zeta_{wNi,j}$ of the new reference tetrahedrons $\Gamma_{i,jn}$ of the described iterative optimization process.

| Iteration Step [No] | $\varphi_{Hw1,1n}$ [°] | $\varphi_{Hw2,1n}$ [°] | $\varphi_{Hw2,2n}$ [°] | $\varphi_{Ht1,1n}$ [°] | $\varphi_{Ht1,2n}$ [°] | $\varphi_{Ht2,1n}$ [°] | $S_{Min}$ |
|---|---|---|---|---|---|---|---|
| 1 | 0.00 | 0.00 | 0.00 | 20.17 | 21.23 | 0.00 | 857.7 |
| 2 | 0.17 | 0.13 | 0.00 | 18.85 | 19.90 | 0.16 | 751.4 |
| 3 | 0.52 | 0.26 | 0.00 | 13.42 | 14.47 | 0.80 | 390.2 |
| 4 | 0.86 | 0.39 | 0.00 | 8.02 | 9.08 | 1.41 | 149.6 |
| 5 | 1.40 | 0.58 | 0.00 | 0.20 | 0.88 | 2.34 | 8.6 |
| 6 | 1.40 | 0.58 | 1.23 | 0.20 | 0.88 | 1.12 | 5.9 |
| 7 | 1.41 | 0.58 | 1.19 | 0.44 | 0.64 | 1.19 | 5.7 |

**Table 10.** Values of components $l_{ki,j}$, $m_{ki,j}$, $m_{ki,j}$ of the unit directional vectors of side edges $k_{i,j}$ of $\Gamma_n$.

| Vertex | $l_{ki,j}$ | $m_{ki,j}$ | $n_{ki,j}$ |
|---|---|---|---|
| $k_{0,0}$ | 0.000 | 0.000 | 1.000 |
| $k_{1,0}$ | 0.000 | 0.240 | 0.971 |
| $k_{0,1}$ | 0.155 | 0.000 | 0.988 |
| $k_{1,1}$ | 0.155 | 0.240 | 0.959 |
| $k_{2,0}$ | 0.000 | 0.549 | 0.836 |
| $k_{2,1}$ | 0.153 | 0.549 | 0.822 |
| $k_{0,2}$ | 0.307 | 0.000 | 0.952 |
| $k_{1,2}$ | 0.308 | 0.239 | 0.921 |
| $k_{2,2}$ | 0.312 | 0.551 | 0.774 |

The directrices of each shell segment of the searched roof structure based on ellipsoid $\sigma$ are sums of sections $w_{i,j}$ of arbitrary ellipses, for instance one directrix corresponding to $w_2$ is the sum of $w_{1,2} = \zeta_{wN1,2n} \cap \sigma$ and $w_{2,2} = \zeta_{wN2,2n} \cap \sigma$ (Figure 31). In addition, $w_{1,0}$ $w_{2,0}$ $w_0$, $w_{1,1} \neq w_1$, $w_{2,1} \neq w_1$, $\{w_{1,1g}, w_{1,1}\}$ $\zeta_{wN1,1n}$, $\{w_{2,1g}, w_{2,1}\}$ $\zeta_{wN2,1n}$, $w_{1,2} \neq w_{2,2} \neq w_2$. The lines $w_{1,1}$ and $w_{1,1g}$ as well as $w_{2,1}$ and $w_{2,1g}$ are coplanar sections of the directrices. The index $g$ denotes that the proper curve e.g. $w_{2,1g}$ is located outside of the reference ellipsoid. It is possible to obtain such a structure that the conditions $w_{1,1} = w_{1,1g} = \zeta_{wN1,1n} \cap \sigma$ and $w_{2,1} = w_{2,1g} = \zeta_{w2,1} \cap \sigma$ are met. The visualization of the resultant architectural form of the discussed structure is presented in Figure 32.

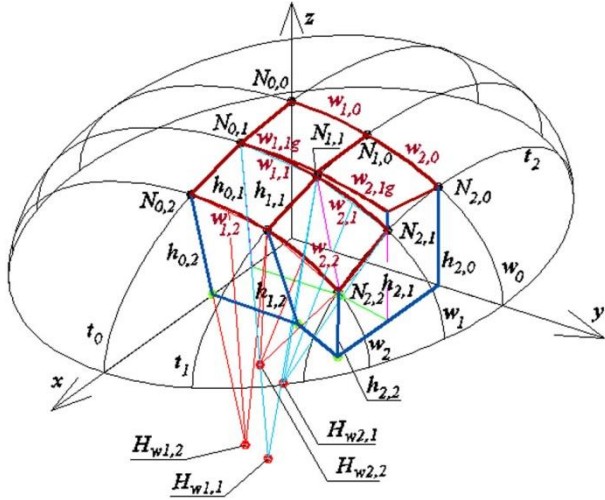

**Figure 31.** One of four parts of the reference structure and roof shell structure symmetrical towards (*x*, *z*) and (*y*, *z*) planes.

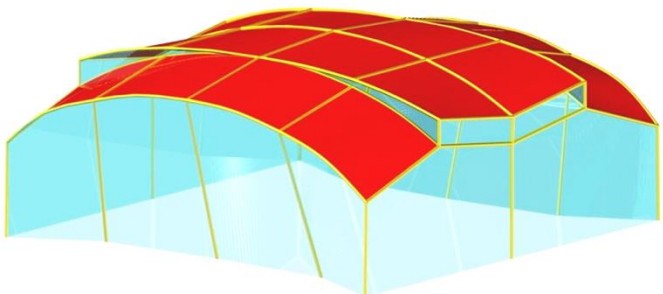

**Figure 32.** Visualization of the optimized free form structure.

## 8. Conclusions

Despite the relatively great possibilities of the search for diverse single free forms of buildings roofed with transformed shells, resulting from the freedom in selecting the shape and position of the roof directrices, there are significant limitations in creating these forms due to the geometrical and mechanical properties of the folded steel sheets. In order to overcome these limitations, the author proposed various methods for shaping the buildings as free form structures composed of many individual forms connected with common walls. Positive e effect of the skillful composition of many single warped surfaces for roofing is that the designed building free form structure is becomes internally consistent and externally sensitive to the built or natural environments. The possibility of further modification of these structures by means of displacements of roof directrices and elevation edges in the planes of the auxiliary reference tetrahedrons, defined by the author, allows the aforementioned internal coherence and external sensitivity to be increased.

Three methods of creating composite building free forms roofed with structures of many shell segments made up of transformed corrugated sheeting are proposed. Based on the results of studies on the first method, it can be concluded that the reference tetrahedrons and operations proposed in the algorithm of this method enable easy and creative creation of such complex free forms characterized by integrated forms of roofs and façades.

Moreover, it is very easy to modify these complete tetrahedrons in order to obtain many different configurations of the free forms sensitive to the natural and built environments. This modification consists of: (1) changing the position of the roof eaves' corners along the side edges of the façade walls in order to change the mutual position of shell roof segments; and (2) changing the position of the

vertices of the reference tetrahedrons along the axes of the tetrahedrons to obtain the corrugation of flat façade walls.

The algorithm of the second method introduces a certain regularity in the placement and joining of subsequent reference tetrahedrons in the three-dimensional space into one regular spatial polyhedral reference network. To achieve this regularity, an auxiliary reference surface is introduced as a double-curved regular surface whose specific properties are used to build and arrange the reference tetrahedrons which are the meshes of the reference network. The algorithm is of no particular support for the designer because it does not offer additional conditions, allowing the form of the reference network to be regular and take into account the variable curvature of the reference surface.

Such additional conditions, effectively supporting the designer's activity, are provided by the very sophisticated third method proposed by the author. The method replaces straight lines normal to the reference surface with side edges of the searched reference network. However, each pair of the adjacent side edges of the reference network must intersect, while the respective two straight lines normal to the reference surface are skewed. Therefore, to solve this problem, the algorithm of the method is based on the optimization of the directions of several side edges of the reference network in relation to a finite number of selected straight lines normal to the reference surface. As a result, the differences in the directions of the side edges and corresponding normals are as small as possible.

The algorithm uses an optimization process, the idea of which is to search for the positions of selected planes of the reference network so that the position of each plane was the closest possible to two subsequent normals to the reference surface. The obtained pairs of subsequent planes have to intersect at the side edges of the reference network that approximate the position of the above normals to the reference surface.

Each plane of the reference network is defined by means of three points. Two of these are points of the intersection of two subsequent normals with the reference surface. The third point is sought on a straight line perpendicular to the above two normals, and intersecting these normals. Therefore, the position of this point is optimized on the aforementioned straight line, for several planes of the reference network. The result of the optimization carried out in the article indicates that the optimal position of each such a point is not, as might be expected, the middle of the section with its ends at the intersecting points of the above three straight lines, but this position is dependent on the changes in curvature on the reference surface and must be calculated during the optimization process.

Obviously, this location is determined by the variability of the curvatures of the reference surface. Therefore, in the future, the author intends to develop a parametric description of the relationship between the overall dimensions and curvatures of the arbitrary smooth regular reference surface and the properties of the optimized reference network searched for the reference surface. In addition, this description should take into account the choice of other characteristic lines on the reference surface, such as geodesic or curvature lines. This description will allow writing a relevant computer application supporting the designer in shaping complex building free forms.

**Funding:** The resources of the Rzeszow University of Technology.

**Conflicts of Interest:** The author declares no conflict of interest.

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
