# Peer review of "Transformed Shell Roof Structures as the Main Determinant in Creative Shaping Building Free Forms Sensitive to Man-Made and Natural Environments"

_buildings, doi:10.3390/buildings9030074_

Round 1

Reviewer 1 Report

Why do authors choose not to cite the following ?

Vizotto, I. (2010). Computational generation of free-form shells in architectural design and civil engineering. Automation in Construction19(8), 1087-1105.

Bletzinger, K. U., Wüchner, R., & Kupzok, A. (2006). Algorithmic treatment of shells and free form-membranes in FSI. In Fluid-structure interaction (pp. 336-355). Springer, Berlin, Heidelberg.

Adriaenssens, S., Ney, L., Bodarwe, E., & Williams, C. (2012). Finding the form of an irregular meshed steel and glass shell based on construction constraints. Journal of Architectural Engineering18(3), 206-213.

Author Response

Thank you for the comments for improvement of this manuscript. I took account of the suggestions. In particular, References are improved.

Reviewer 2 Report

The paper presents 3 methods for shaping free forms of buildings sensitive to harmonious incorporation into the built environment. These methods are compared. The introduction presents the problems linked to this topic, and a literature review is intrusive in the next paragraph. Could you compare your methodology with others? 

The methodology of you paper is not clear. Could you introduce a paragraph on it? 

Other parts are more a report than a paper. The structure and the description of results must be improved and clarified. Could you applicate your theory in a case study? No results are presented. The paper must be improved inserting the results and the comparison among the 3 methodologies and other methodologies. 

The conclusion are interesting but not referred to the results. 

Author Response

Thank you for the sets of critical reviews and comments for technical improvement of this manuscript. The summary of my specific response is as follows.

Currently, my basic scientific activity is focused on shaping free forms of buildings, in particular free form structures roofed with many transformed shell segments. These structures are characterized by a complex free form of flat-walled folded façades, and are roofed with an integrated complex shell structure of several complete shells. The basic purpose of my manuscript is to present new possibilities of shaping these structures using three methods that differ in the complexity of algorithms and regularities, and which allow obtaining structures that differ in qualitative rather than quantitative properties of the free form. The methods, presented in such a proposed order, are increasingly sophisticated in the creative search for coherent forms of the complex free buildings sensitive to the natural or built environments.

Only after I have completed a coherent step of my research on the aforementioned topics, I intend to continue more comprehensive innovative studies on the geometric and static-strength properties of complete shells made of transformed folded steel sheets, and unconventional constructions dedicated to such thin-walled corrugated roof shells. So far I have published only preliminary results of my research in this area (Figure 1 of my manuscript). I have pointed out the need for more in-depth, detailed experimental tests, analyses and computer models, also in reference to the works of the authors referred to in Sections 3 – 4 and References. Therefore, in this article, I am not able to compare at length and indicate the quantitative differences (which was not my aim in the first place) in the results of my work and those of the aforementioned authors.

I participated in some of the studies done by Prof. Reichhart, whose selected samples of transformed folded shells I presented in Figure 2 of my manuscript. My participation was limited to geometric shaping of transformed corrugated shell sheeting obtained in the laboratory conditions in the university hall. In Figure 3, I have shown some of the experimental corrugated shells I studied, the results thereof I published in my doctoral dissertation and the monograph.

Prof. Reichhart has retired and no longer conducts the scientific research and engineering activity. I try to continue the problems he began, so I started with my research almost from the beginning, and none the free form buildings roofed with a transformed shell of my design have actually been built. Recently, I have designed an innovative steel construction for experimental tests in the field of geometric and static-strength properties of thin-walled transformed corrugated steel shells. The construction has already been made in the workshop and will be assembled in the coming weeks - using bolted connections in the university's laboratory hall.

In the experimental tests planned by myself, this universal construction will provide supporting conditions for variously deformed, of various profiles subsequent folds of the sheeting of the transformed shell, according to the conditions on the construction site and consistent with the designer’ expectations. My innovative solutions enable modification of the support of the subsequent folds to skew roof directrices in relation to their expected shell forms, width and inclination to the horizontal base plane. These tests will allow me to accurately configure my initial, accurate, thin-walled computer models of the transformed folded sheets made in the ADINA program (Figure 1) used for non-linear, dynamic, static and strength analyses of members and structural systems of buildings.

I recently presented a method of shaping single building free forms roofed with transformed roofs in one of the articles published in Buildings, in which my concepts were supported by the co-author’s experience in architecture. In my manuscript currently submitted for review, I analyze the possibilities of shaping complex forms of buildings roofed with structures of several transformed shell segments. In this manuscript I have made only a qualitative comparison between my three methods, and between these methods and the methods proposed by other authors. This is because other authors propose a rather limited range of the variety of forms of the structures, which are called hypars (that is only roof shells but not entire buildings). They propose only a few different, very specific configurations of the structures composed of one type of segment, i.e. the central segment of the right hyperbolic paraboloid or its quarter (its one-fourth).

Of the three innovative methods I propose, the first one is the most general and based on a small number of simple rules. That is why it is proposed for the search for both regular and irregular forms composed of shell roof structures and entire complex buildings with flat-walled folded facades. The other two methods I propose concern problems of transformed, rectangular folded sheets, where the forms of transformed shells are determined by orthotropic geometrical and mechanical properties of the sheets, which problems have not hitherto been investigated by the aforementioned researchers.

Especially in the third, most complex and sophisticated method, I propose to create a regular polyhedral network composed of many regular specific tetrahedrons, such that the position of their side edges is optimized in relation to a finite number of selected straight normals to almost any double curved auxiliary regular surface called the reference surface. The proposed rules, objects and activities ensure the regularity of the roof structure, integration of the structure with the folded façade form and allow the free form building to be adapted to the natural or built environments.

Only in the near future, after completing the intended research including experimental tests, the above three methods will be supplemented with accurate data allowing for computer aided parametric shaping of free form structures, a parametric description of their static-strength work and structures dedicated to these forms. That is why I am currently unable to meet the essential postulates of the reviewer. I will do it after obtaining satisfactory results of the laboratory tests I have planned on the research stand of my design that will be assembled in the university laboratory hall.

I have essentially modified the abstract, entire chapters 3 and 4 and the introductory part of chapter 5, in an attempt to precisely describe the research methodology I have adopted, the results thereof I described in Sections 5 – 7. Despite all the effort, I was not able to change the structure of my manuscript because of the goals and scope of my research and the publication of results I had adopted. I have tried, however, to the best of my ability, to make the presentation of the goals, methodology and ways of solving the problems clearer and more precise.

Therefore I have made the aforementioned changes and significantly modified the content of Section 7 presenting my most advanced and sophisticated method for creative shaping of building structures roofed with structures composed of many regular shell segments. I received the Tsuboi Award for the best article in the Journal of the International Association for Shell and Spatial Structures, in one of the previous years, for the article presenting the content that is the introduction to the concept of the third method presented in the submitted manuscript. The award is the result of my initial scientific research. In Section 7, I present the coherent complete method and process used to optimize the free form structures of many shell sectors, which allows practical use in shaping regular polyhedral networks, shell roof forms and complex free forms of entire buildings. I have also made the conclusions more precise so that they accurately interpret the results obtained by means of the three methods I propose.

Round 2

Reviewer 1 Report

The authors have improved the manuscript

Reviewer 2 Report

ok